**RESEARCH**

# CRISPR screens identify gene targets at breast cancer risk loci

Natasha K. Tuano[1], Jonathan Beesley[2], Murray Manning[1,3], Wei Shi[2], Laura Perlaza-Jimenez[1,4], Luis F. Malaver-Ortega[3], Jacob M. Paynter[1,5], Debra Black[2], Andrew Civitarese[2], Karen McCue[2], Aaron Hatzipantelis[2], Kristine Hillman[2], Susanne Kaufmann[2], Haran Sivakumaran[2], Jose M. Polo[5], Roger R. Reddel[6], Vimla Band[7], Juliet D. French[2], Stacey L. Edwards[2], David R. Powell[4], Georgia Chenevix-Trench[2]* and Joseph Rosenbluh[1,3]*

*Correspondence:
Georgia.Trench@qimrberghofer.
edu.au; sefi.rosenbluh@monash.
edu

[1] Cancer Research Program
and Department of Biochemistry
and Molecular Biology,
Biomedicine Discovery Institute,
Monash University, Clayton, VIC,
Australia
[2] Cancer Program, QIMR
Berghofer Medical Research
Institute, Brisbane, Australia
Full list of author information is
available at the end of the article

## Abstract

**Background:** Genome-wide association studies (GWAS) have identified > 200 loci associated with breast cancer risk. The majority of candidate causal variants are in non-coding regions and likely modulate cancer risk by regulating gene expression. However, pinpointing the exact target of the association, and identifying the phenotype it mediates, is a major challenge in the interpretation and translation of GWAS.

**Results:** Here, we show that pooled CRISPR screens are highly effective at identifying GWAS target genes and defining the cancer phenotypes they mediate. Following CRISPR mediated gene activation or suppression, we measure proliferation in 2D, 3D, and in immune-deficient mice, as well as the effect on DNA repair. We perform 60 CRISPR screens and identify 20 genes predicted with high confidence to be GWAS targets that promote cancer by driving proliferation or modulating the DNA damage response in breast cells. We validate the regulation of a subset of these genes by breast cancer risk variants.

**Conclusions:** We demonstrate that phenotypic CRISPR screens can accurately pinpoint the gene target of a risk locus. In addition to defining gene targets of risk loci associated with increased breast cancer risk, we provide a platform for identifying gene targets and phenotypes mediated by risk variants.

**Keywords:** Post GWAS, Breast cancer risk, Functional phenotypic screens, Target discovery

## Background

Genetic evidence that implicates a gene in disease etiology is a strong indicator that drugs targeting the encoded protein will be effective as therapies or for risk reduction [1, 2]. Indeed, one of the most commonly used drugs for primary and secondary prevention of BC is tamoxifen, an antagonist of the estrogen receptor (ER) which is

encoded by *ESR1*, a known target of a BC risk locus [1]. GWAS have identified > 200 loci associated with BC risk, most of which are associated with both ER + and ER − BC [2]. These loci represent a valuable source for identifying drug targets [3–5], but translation of these findings to actionable mechanisms requires first identifying the target gene of the association. However, since most CCVs are in non-coding regions (Fig. 1A), and are presumed to act through regulatory mechanisms, identification of the target gene is challenging.

Current strategies to identify gene targets of non-coding GWAS variants frequently rely on functional genomics assays such as high-throughput chromatin interaction capture [6, 7] or the recently described CRISPRqtl [8]. These approaches require the use of cultured cell lines or small quantities of primary tissue material to infer target genes in disease-relevant samples. Since relevant intact tissue is often not available, and given the dynamic nature of cell- and context-specific transcriptional enhancer activity, it is challenging to identify targets of non-coding variants. Furthermore, enhancers typically regulate the expression of multiple genes ("molecular pleiotropy") [9, 10], and thus, even if enhancer activity is preserved, functional genomics assays may identify multiple candidate genes and additional experiments are needed to define the causal genes.

A complementary approach is to use a phenotypic readout to identify putative GWAS target genes which mediate a cancer phenotype and then use functional genomics to evaluate their regulation at GWAS loci. We hypothesize that genes implicated by GWAS, with strong in silico supporting evidence, will influence a quantifiable cancer phenotype which will enable us to nominate the most likely BC risk genes. Pooled CRISPR screens are extensively used to identify genes related to a particular phenotype [11, 12] but have not been used to characterize GWAS target genes.

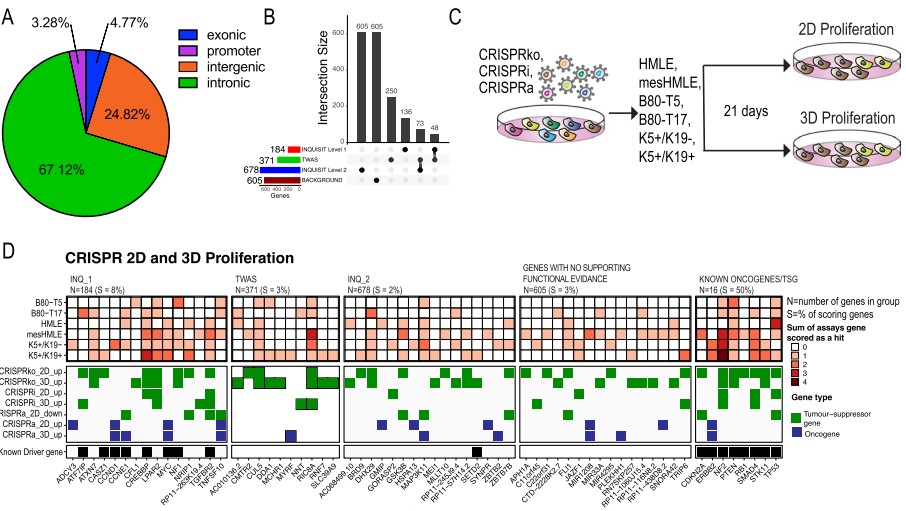

**Fig. 1** CRISPR activation and suppression screens identify BC risk genes that regulate proliferation in 2D, 3D. **A** Pie chart showing locations of CCVs at 205 BC risk signals identified by GWAS. **B** Classes of genes selected for functional CRISPR screens. INQ_1 – high-confidence INQUISIT predictions; INQ_2—moderate-confidence INQUISIT predictions; TWAS – identified by transcriptome wide association studies and eQTL studies. **C** Experimental approach. **D** Summary of results from 2 and 3D proliferation screens. Known driver gene annotations were from [3]

In CRISPR screens, the target gene is directly inhibited or activated, and thus, it is not necessary to have a cell line model with a preserved enhancer structure that is reflective of the relevant primary cells. Furthermore, CRISPR screens define a phenotype and directionality, thereby generating a cell model that could be used for further mechanistic studies and development of inhibitors.

We recently developed a heuristic gene scoring system called INQUISIT (Integrated expression quantitative trait and in silico prediction of GWAS targets) to rank the predicted target genes at BC risk loci [3, 4]. INQUISIT treats any CCV as potentially able to influence the regulation of distal genes (via chromatin looping from enhancers), proximal genes through promoter modulation, and as consequences of coding and splicing changes (See example of INQUISIT in Additional file 1: Fig. S1). We designated INQUISIT predictions with the strongest supporting evidence as level 1, and level 3 as the lowest. At 205 fine-mapped risk signals (having omitted one with > 2000 CCVs), INQUISIT identifies 1–10 level 1 targets per signal at 114 signals (184 genes). For 76 signals, INQUISIT predicts 678 level 2 genes, and for 15 signals, INQUISIT does not predict any gene targets (Additional file 2: Table S1).

However, computational algorithms can only predict candidate target genes for a particular locus. Pinpointing the gene target and identifying the phenotypes it mediates requires experimental data, which is a daunting task for so many BC risk loci. Here, we show that large-scale pooled CRISPR activation and suppression phenotypic screens are highly effective in identifying genes which mediate proliferation in vitro, tumor formation in vivo and DNA damage response, in order to define gene targets at BC risk loci.

## Results

### Selection of genes for functional CRISPR screens

We selected genes using the following approaches: (1) 184 high-confidence INQUISIT level 1 (INQ_1) target genes [3], (2) 678 INQUISIT level 2 (INQ_2) target genes, (3) 371 genes identified by Transcriptome Wide Association Studies (TWAS) and expression quantitative trait loci (eQTL) studies of BC risk (collectively referred to as TWAS genes) [9–12], (4) 605 "genes with no supporting functional evidence" (background genes), which include low confidence INQUISIT targets, genes predicted by INQUISIT prior to fine mapping, and genes predicted only in an early version of INQUISIT, as well as genes located within 2 Mb of 15 risk signals at which INQUISIT did not predict any targets. Genes predicted by both INQUISIT and TWAS were categorized as INQ_1 for level 1 predictions and as TWAS for level 2 predictions (Fig. 1B and Additional file 2: Table S1). For each of these genes, we designed five single guide (sg)RNAs. In addition, we included 1000 negative controls and 960 sgRNAs targeting 193 core essential genes, as well as 16 known tumor-suppressor genes and oncogenes (Additional file 3: Table S2).

### BC risk genes that induce a proliferation phenotype in 2D and 3D cultures

Impaired proliferation is a hallmark of cancer [13]. We used CRISPR screens, in six immortalized mammary epithelial cell lines, to suppress (CRISPRko or CRISPRi) or overexpress (CRISPRa) candidate BC-risk genes and identify putative tumor-suppressors and oncogenes (Fig. 1C). For CRISPRi and CRISPRa screens, we selected sgRNAs targeting the gene promoter, and for CRISPRko screens, we selected sgRNA

targeting the first exons as previously described [14–16]. Expression and ATAC-Seq profiling indicated that these six cell lines represent breast cells with either a luminal progenitor signature (K5 + /K19 + [17], K5 + /K19 − [17]), a mesenchymal signature (B80-T17 [18], mesHMLE [19]), or a more epithelial like signature (B80-T5 [18], HMLE [20]) (Additional file 1: Fig. S Fig. S2A-C). Using this dataset, we found that, as expected, INQ_1 and TWAS genes which are selected also based on having high gene expression had consistently higher levels of gene expression compared to genes with no functional evidence (Additional file 1: Fig. S2D,E). Following library infection, cells were plated in 2D or 3D conditions and sgRNA abundance was quantified after 21 days (Additional file 4: Table S3). Negative controls had no proliferation effect and as expected, suppression of core-essential genes had a negative impact in CRISPRko and CRISPRi screens but no effect in CRISPRa screens (Additional file 1: Fig. S3A). Known tumor-suppressors had a positive proliferation impact in CRISPRko and CRISPRi screens (Additional file 1: Fig. S3A). Although some known oncogenes showed increased proliferation in CRISPRa screens (e.g., *ERBB2*), we did not see overall increase in proliferation most likely because many of the known oncogenes we selected are context specific and do not show a proliferation phenotype in breast cells (e.g., *YAP1*, *KRAS*). Using the MAGeCK algorithm [21], we compared sgRNA abundance after 21 days of growth to sgRNA abundance in the DNA pool and calculated for each gene the Log2[FoldChange] and FDR (Additional file 5: Table S4).

We set the threshold for functional genes with a magnitude of effect Log2[Fold Change] > 1 and a significance -Log10[FDR] > 1 in at least one cell line. Oncogenes were defined as genes that upon overexpression increased proliferation in 2D or 3D cultures. Tumor suppressors were defined as genes that increase proliferation upon suppression in 2D or 3D cultures or genes that inhibit proliferation in 2D cultures (Log2[Fold Change] < -1) upon overexpression. We only used suppression of proliferation as a criterion in 2D and not 3D cultures, since we used immortalized cells that are not able to proliferate in 3D without an additional oncogenic insult. Thus, the proliferation inhibition phenotype in 3D for both negative controls and TSGs look the same. Importantly, we only used overexpression to further support a gene as a tumor-suppressor and not as a stand-alone criterion (Fig. 1D and Additional file 1: Fig. S3B-G).

We identified 41 candidate BC-risk genes that mediate a proliferation phenotype in 2D or 3D cultures (Fig. 1D and Additional file 1: Fig. S3B-G). Our results demonstrate the utility of using multiple assays, cell lines, and perturbation methods. We found high consistency between CRISPRi and CRISPRko screens (Additional file 1: Fig. S4A). The inconsistencies detected are likely due to CRISPRi bidirectional promoter off-target effects [16, 22]. For example, *ATXN7* scored as a strong tumor-suppressor using CRISPRko in 2D and 3D assays but did not score with CRISPRi (Additional file 1: Fig. S4B). This is because *ATXN7* and *THOC7* share a bidirectional promoter (Additional file 1: Fig. S4C) and *THOC7* is a common cell essential gene (Additional file 1: Fig. S4D, E). Thus, CRISPRi sgRNAs targeting the *ATXN7* promoter also inhibit *THOC7*, resulting in cell death. We found high correlation between 2 and 3D proliferation changes (Additional file 1: Fig. S5A-C). Interestingly, some genes showed the opposite effect in 2D and 3D cultures suggesting a function

in mediating cell motility. For example, *CFL1* scored as a potent tumor-suppressor in 3D cultures but had no effect in 2D cultures (Additional file 1: Fig. S5D). This is consistent with the known function of *CFL1* as a regulator of actin filament polymerization and cell motility [23, 24].

### Validation of 2D and 3D proliferation hits

To validate these observations in a singleton experiment, we infected all six cell lines with individual sgRNAs targeting INQUISIT level 1 hits that scored in CRISPRko or CRISPRa screens. Using western blot or qRT-PCR, we confirmed that these sgRNAs reduced (for tumor-suppressor genes) or increased (for oncogenes) expression of the target protein (Additional file 1: Fig. S6A, B). Following infection, cells were plated on 2D (Additional file 1: Fig. S7A, B) or 3D (Additional file 1: Fig. S7C, D) conditions and proliferation was measured by monitoring cell growth. Consistent with our screening results, we were able to validate the 2D and 3D proliferation effects in at least one cell line. For *CREBBP* and *CFL1*, we found a statistically significant and robust (effect > 20%) cell line-specific effect on 2D proliferation with CRISPRko (Additional file 1: Fig. S7A, B). This is consistent with reports showing that *CREBBP* can act as a tumor-suppressor or an oncogene in a cell type specific manner [25, 26].

In summary, our approach identified 41 candidate BC risk genes that mediate a proliferation phenotype in 2D or 3D cultures, including 15 that were predicted with high confidence by INQUISIT. We validated the gene knockout or overexpression and phenotypes for 14 of these genes. These genes include well annotated tumor-suppressor genes and oncogenes (e.g., *TGFBR2* and *MYC* as well as genes that have never been previously linked to cancer in general, or to increased BC risk (including *ADCY3, ATXN7, CFL1* and *LPAR2*). Together, these results demonstrate the ability of systematic CRISPR screens to define genes associated with BC risk that drive a proliferation phenotype.

### Identification of BC-risk genes that promote tumor formation in mice

To identify candidate BC-risk genes that play a role in tumor formation in vivo, we used a mouse xenograft model (Fig. 2A). To enable tumor formation in mice, we found that the above-described cell lines required an additional, constitutively active form of MEK1 (MEKDD) in addition to a CRISPR-mediated oncogenic insult (Additional file 1: Fig. S8A). Following library transduction, cells were injected to the flanks of immune deficient mice. Tumors were harvested 6–8 weeks post injection, and sgRNA abundance was quantified (Additional file 1: Fig. S8B-D and Tables S3, S4). Five positive controls showed dramatically increased sgRNA abundance and ten INQUISIT-predicted genes (five at level 1) scored in this assay, suggesting these are potent drivers of BC-risk (Fig. 2B). We validated INQUISIT level 1 CRISPRko hits in B80-T5-MEKDD cells (Additional file 1: Fig. S8E, F).

We found that *DUSP4, CTD-2278I10.4,* and *VPS45* scored only in vivo and had no effect in vitro. *DUSP4* is an INQUISIT level 1 hit which we have previously shown to be down-regulated by CCVs at 8p12 [27]. Since our in vivo screens used MEKDD expressing cells, we explored whether *DUSP4* is a context-dependent tumor-suppressor. Following *DUSP4* knockout in B80-T5 or B80-T5-MEKDD, we measured proliferation in 3D cultures. Consistent with the in vivo screen, *DUSP4* knockout

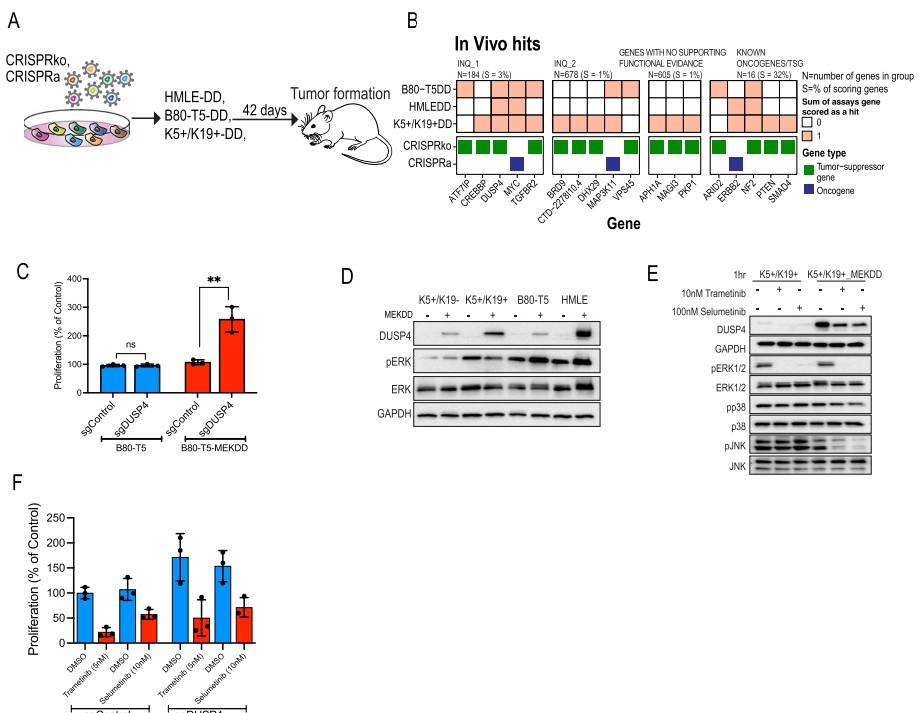

**Fig. 2** CRISPR activation and suppression screens identify BC risk genes that regulate tumor growth in mice. **A** Experimental approach. **B** Summary of hits from the in vivo screens. **C** Proliferation in 3D cultures of WT or MEKDD expressing B80-T5 cells following suppression of *DUSP4* expression. **D** DUSP4 protein levels following expression of MEKDD. **E** Phosphorylation of ERK, JNK and p38 following MEK inhibition. **F** The effect of MEK inhibitors on proliferation in *DUSP4* ko cells

only showed increased proliferation in the presence of MEKDD (Fig. 2C). To investigate *DUSP4's* mechanism of action, we assessed DUSP4 levels following MEKDD expression and found that MEKDD induced DUSP4 expression, suggesting a negative feedback loop (Fig. 2D). However, consistent with previous results [28], we did not observe any change in pERK levels following suppression of *DUSP4* (Additional file 1: Fig. S9A, B). We did observe decreased pJNK and pp38 in *DUSP4* knockouts (Additional file 1: Fig. S9A, B), confirming previous observations [29, 30] and suggesting that downregulation of pJNK via *DUSP4* mediates its tumor suppressive activities. To further validate these observations, we used selumetinib and trametinib, two potent MEK inhibitors. We found that MEK inhibitors reversed the increased DUSP4 protein levels (Fig. 2E and Additional file 1: Fig. S9C) as well as *DUSP4* induced proliferation (Fig. 2F) suggesting MEK inhibitors as a therapeutic strategy in BC with downregulated *DUSP4* expression. This might be particularly relevant in triple negative BC because *DUSP4* is deleted in about 50% BC, most often in this aggressive subtype [30, 31].

Together, our in vivo and in vitro proliferation screens identify 44 predicted BC risk genes (including 16 INQUISIT level 1 genes) that can drive a proliferation phenotype in 2D, 3D cultures or in vivo (Figs. 1D and 2B). We found a strong correlation in phenotypes between the different cell lines (Additional file 1: Fig. S9D), indicating that even if a particular gene did not pass our threshold, it is likely to be a near hit in other cell lines.

### Identification of genes that regulate the DNA damage response

DNA damage is a hallmark of cancer in general and in particular is deregulated in BC [32]. To identify BC risk-associated genes that regulate the DNA damage response, we used a PARP inhibitor synthetic lethality screen [33]. Following sgRNA infection, cells are treated with olaparib, a potent PARP1 inhibitor, and only cells harboring a sgRNA that deregulates the homologous recombination DNA damage repair pathway are rendered sensitive to olaparib (Fig. 3A, B). By comparing sgRNA abundance in olaparib- and DMSO-treated cells, we identify genes that upon suppression or overexpression sensitize cells to olaparib. Using this approach, we screened 5 immortalized mammary cell lines for genes that upon suppression (CRISPRko or CRISPRi) or overexpression (CRISPRa) regulate the DNA repair pathway (Fig. 3C and Additional file 1: Fig. S10A-F). We identified 27 candidate BC risk genes that regulate the DNA damage response, including 8 that were predicted with high confidence by INQUISIT. As expected, Gene Set Enrichment Analysis (GSEA) of hits (not including positive controls and background genes) showed enrichment for genes involved in the DNA repair pathway and cell cycle (Fig. 3D). This is consistent with the two types of cellular stresses known to be synthetic lethal with mutations in the DNA repair pathway [33]. Indeed, four of the INQUISIT level 1 scoring genes (*MYC, NF1, NRIP1,* and *CREBBP*) also scored in the

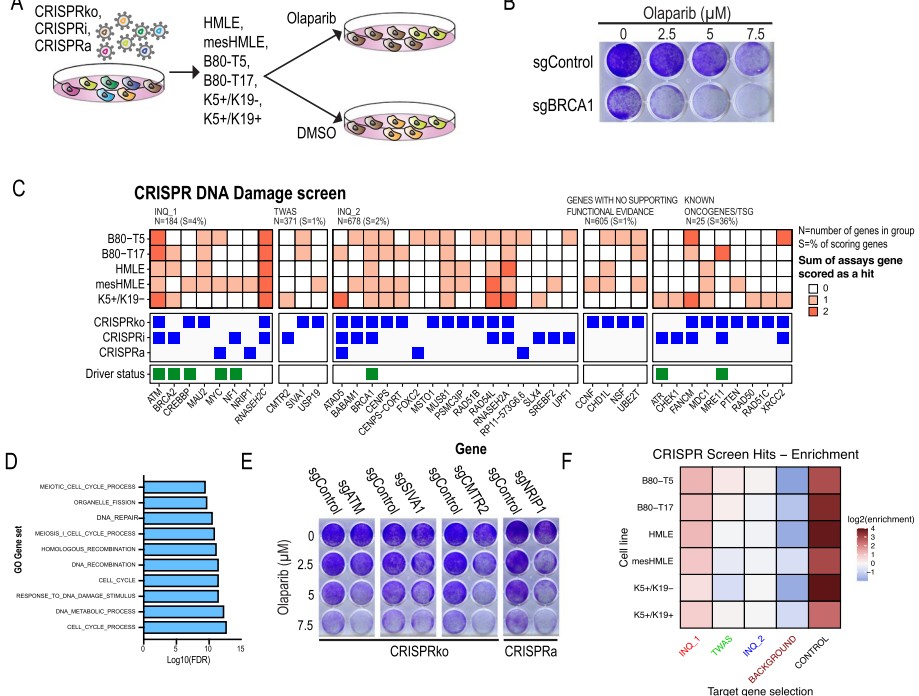

**Fig. 3** Olaparib synthetic lethal screens identify BC-risk genes that regulate the DNA repair pathway. **A** Experimental approach. **B** B80-T5 cells infected with control or *BRCA1*-targeting sgRNAs were treated with olaparib for 7 days. **C** Summary of hits. Known DNA repair genes were annotated based on [33]. **D** GSEA pathway enrichment analysis of hits (not including background positive control genes). **E** Validation of selected hits in a singleton experiment using a crystal violet readout. **F** Enrichment of genes predicted by various computational and statistical methods among the combined hits from all phenotypic screens. INQ_1—high-confidence INQUISIT predictions (genes scored as INQ_1 due to their driver status were downgraded to INQ_2 for this analysis); INQ_2—moderate-confidence INQUISIT predictions; TWAS—identified by transcriptome wide association studies or eQTL studies

above-described proliferation screens (Fig. 1D). To validate these results in a singleton experiment, we used B80-T5 cells and showed that CRISPR mediated suppression of known DNA damage-related genes (*ATM*), as well as newly identified synthetic lethal genes (*SIVA1* and *CMTR2*), had a dramatic effect on PARP inhibitor sensitivity (Fig. 3E and Additional file 1: Fig. S10G). In total, including controls, we identified 50 genes that play a role in the DNA damage response. Of these 33 (66%) have been also found in other genome wide DNA damage CRISPR screens [33, 34] (Additional file 1: Fig. S10H). Including positive controls, we identified 17 genes, including three INQUISIT level 1 genes that have not been found in previous large-scale DNA damage CRISPR screens. These experiments define a set of 27 likely BC risk genes that drive a DNA repair phenotype, including eight INQUISIT level 1 genes.

We tested for the enrichment of genes predicted by various computational and statistical methods among the combined hits from all phenotypic screens. We observed an overall enrichment of INQUISIT level 1 genes (1.7-fold, Fisher's exact test $p = 0.04$), with significant over-representation in five of the six cell lines (Fig. 3F). Importantly, the observed enrichment was not confounded by a gene's status as a known driver (a source of up-weighting in the INQUISIT pipeline), since the enrichment was observed even when the genes which were nominated as INQUISIT level 1 on the basis of being known BC driver genes (*CASZ1*, *CCNE1*, *CREBBP*, and *NF1*) were downgraded to INQUISIT level 2. Although genes with no functional evidence had lower expression levels (Additional file 1: Fig. S2D,E), the proportion of genes scoring in this group was similar to TWAS and INQ_2 which had expression levels that are similar to INQ_1 genes. This suggests that high-confidence INQUISIT predictions represent probable candidate genes at disease-associated loci. Taken together, these experiments define a set of 20 INQUISIT level 1 likely BC risk genes, and an additional 46 genes that drive a proliferation or DDR phenotype in breast cells.

### HiChIP and CRISPRqtl validate distal regulation between BC risk loci and genes that score in functional screens

Using CRISPR phenotypic screens we identified 20 genes that are associated with distal enhancers containing BC risk variants. A common mechanism of distal enhancers is to regulate gene expression through chromatin looping [3, 35]. To confirm this in the normal breast cell lines we used in the current screens, we performed HiChIP on B80-T5 and K5+/K19+cells. For 18 of the 20 INQUISIT level 1 hits identified in the screens, we found chromatin interactions with regions containing BC risk variants (Fig. 4A and Additional file 6: Table S5). One of the genes for which we did not identify chromatin interactions was *BRCA2*, which contains a coding CCV. For *NF1* and *RP11263K19.4*, we did not find any looping with their predicted risk region in these cell lines. However, 3D genome structure may not always be consistent between cultured cell lines and intact tissue which demonstrates the added value of phenotypic screens that do not require the same 3D genome structure as in the relevant primary mammary epithelial cells.

The chromatin interactions we found were particularly strong for *ATF7IP* (Fig. 4A). The risk signal at this locus comprises 18 SNPs, seven of which lie within a candidate enhancer region marked by open chromatin and H3K27ac histone marks (Fig. 4B). We used luciferase reporter assays to test whether variants within these enhancers

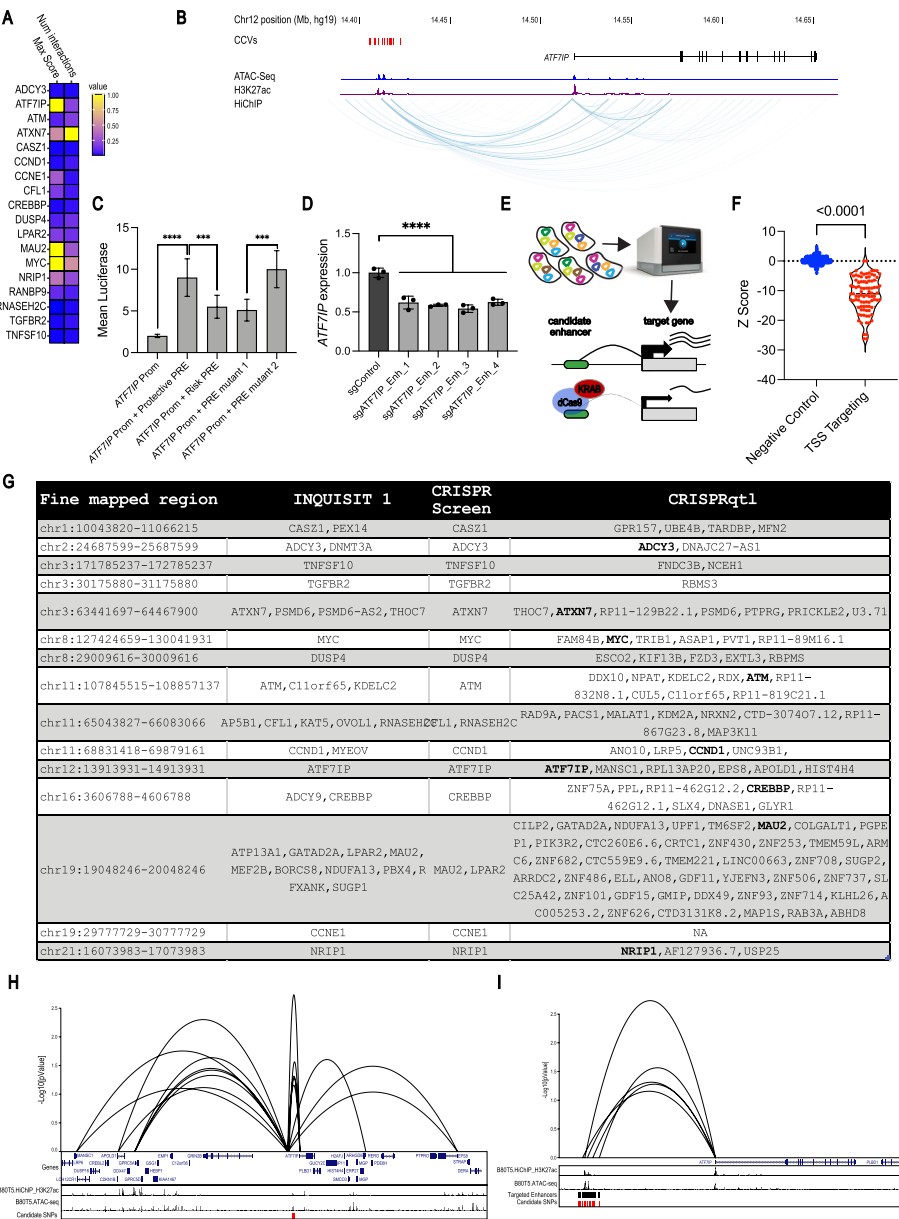

**Fig. 4** Chromatin conformation assays and CRISPRqtl confirm interactions between BC risk loci and genes that score in functional screens. **A** Summary of HiChIP chromatin interactions observed between BC risk loci and genes that scored in functional screens, where color scale signifies scaled levels of chromatin interaction scores and count. **B** Example of chromatin interactions between CCVs and *ATF7IP*. **C** The regulatory element carrying the protective alleles of CCVs rs16909788, rs17221259, rs11055880 increase *ATF7IP* promoter activity. Constructs containing all three SNPs were tested using luciferase reporter assays. PRE mutant 1 contains the protective haplotype with rs11055880 altered to the risk allele. PRE mutant 2 contains the risk haplotype with rs16909788 and rs17221259 altered to protective alleles. Bars show mean luciferase intensity relative to promoter activity and error bars represent 95% confidence intervals. *P*-values were determined by two-way ANOVA followed by Dunnett's multiple comparisons test (****$p < 0.0001$). **D** *ATF7IP* expression was measured in K5 + /K19 − cells 21 days post infection with CRISPRi sgRNAs targeting the *ATF7IP* CCV-containing enhancer. **E** Strategy used for CRISPRqtl experiment. **F** Z-Scores from CRISPRqtl screen of sgRNAs targeting 50 known TSSs. **G** Genes identified by CRISPRqtl screen as targets of 16 fine mapped BC risk regions that had an INQUISIT level 1 and a CRISPR functional screens hit. **H** Example of CRISPRqtl results at the chr12:1,391,331–14,913,931. **I** Zoom into *ATF7IP*, showing 5 enhancers that score in CRISPRqtl as regulators of *ATF7IP* expression

alter *ATF7IP* promoter activity. Addition of the *ATF7IP* putative regulatory element (PRE) containing the protective allele to the *ATF7IP* promoter had a ninefold increase ($p < 0.0001$) in luciferase activity (Fig. 4C). This increase in luciferase activity was reduced by 50% ($p < 0.001$) following introduction of the PRE containing the risk-associated allele. Furthermore, we found that introduction of a variant at rs11055880 (PRE mutant 1) had the same effect as the entire risk associated allele while introduction of rs16909788 and rs17221259 had no effect (PRE mutant 2) on luciferase activity (Fig. 4C). Overall, this effect is consistent with BC risk-associated variation at this locus reducing expression of the putative tumor-suppressor gene, *ATF7IP*. Since luciferase assays require expression of an exogenous construct and may not fully recapitulate the native chromosome structure, we validated these results using a CRISPRi approach. Previous studies showed that sgRNAs targeting enhancers are effective in suppressing the expression of the target gene [36]. Using four *ATF7IP* enhancer-targeting sgRNAs, we found a 50% reduction in *ATF7IP* expression (Fig. 4D), further demonstrating this BC-associated enhancer as a regulator of *ATF7IP* expression.

Based on these results, we performed a systematic CRISPRi enhancer screen using the recently described CRISPRqtl approach [8]. In CRISPRqtl, a pooled sgRNA library targeting putative enhancers is cloned in a vector that is compatible with single cell RNA-Seq (scRNA-Seq). Following transduction at high multiplicity of Infection, scRNA-Seq is used to detect sgRNA identity and global mRNA abundance. All cells expressing a particular sgRNA are aggregated and the sgRNA effect on expression of genes in cis (2 Mbp from the sgRNA) is calculated (Fig. 4E). We used CRISPRqtl to define gene targets of 16 risk loci, containing 18 INQUISIT-1 predicted hits from the functional CRISPR screens (we omitted the three loci that did not show any chromatin interactions in HiChIP (Fig. 4A)). To define candidate enhancer regions, we overlapped ATAC-seq and H3K27ac interaction peaks detected in B80-T5 cells. From the center of these peaks, we expanded 500 bp in both directions to get 1 kb regions. We then intersected these regions with BC risk CCVs identified by fine mapping [3]. Since CCVs are often close to each other, in some cases, these 1 kb regions overlap and are merged. For example, within the fine mapped region on chr21 (chr21:16,073,983–17,073,983), we found 12 candidate enhancers. Eleven of these candidate enhancers had 1–2 CCV, but one candidate enhancer (chr21:16,350,528–16,354,781) had 10 closely spaced CCVs resulting in a 4.2-kb candidate enhancer (Additional file 7: Table S6). For each candidate enhancer, we used the CRISPRpick [14] algorithm to design an enhancer tilling sgRNA library (1 sgRNA every 100 bp) (Additional file 7: Table S6). In addition, we included 50 negative controls (non-targeting sgRNAs) and 100 sgRNAs targeting the TSS of 50 genes from [8] as positive controls (Additional file 3: Table S2).

Following transduction B80-T5 cells at MOI $= 5$, three lanes of $10 \times$ chromium were used to collect single cells, and cDNA generated from these cells was sequenced. The CRISPR application in the Cell Ranger package [37] was used for deconvolution and alignment to the human genome. We detected a total of 24,490 cells with a mean of 49,515 reads/cell and a median unique molecular identifier (UMI) of 15,186 UMIs/cell. To reduce non-specific noise due to low level sgRNA detection [38], we filtered out cells that did not have a minimum of two sgRNA UMIs/cell and at least three reads/sgRNA UMI. We also filtered out cells with high mitochondrial content and cells with $< 200$

genes detected resulting in 13,667 cells. To identify genes regulated by these enhancers, we used the recently described SCEPTRE algorithm [39]. SCEPTRE uses conditional resampling and avoids confounding issues associated with high-throughput expression profiling experiments. For each candidate enhancer, we grouped all targeting sgRNAs and calculated the effect (*Z*-score and *p*-value) of CRISPRi targeting of the candidate enhancer on expression of genes in a 2-Mbp window from the enhancer (Additional file 7: Table S6).

As expected, and consistent with previous reports [8], TSS targeting sgRNAs had a dramatic effect ($p < 0.0001$) on expression of their target gene (Fig. 4F), demonstrating the reliability of this approach. As predicted target genes, we considered sgRNA-gene pairs that showed a *p*-value $\leq 0.1$ and a negative *z*-score. Using these criteria, we found 113 genes regulated by 15 fine mapped BC-associated risk signals (Additional file 7: Table S6). For one region (chr19:29,777,729–30,777,729), we did not find any significant hits. For most of these enhancers, CRISPRqtl identified multiple gene targets (e.g., 28 gene targets for chr19:19,048,246–20,048,246) most likely due to molecular pleiotropy. As a CRISPRqtl example, we show sgRNA-gene pairs at the BC-risk locus at chr12:13,913,931–14,913,931. Consistent with our functional screen, HiChIP and luciferase assays, CRISPRqtl also found a strong interaction between this locus and *ATF7IP* (Fig. 4H,I), demonstrating the value of using a multi-assay approach to define GWAS targets.

In total, of the 21 INQUISIT level 1 genes that scored in our phenotypic CRISPR screens, we found an overlap with 18 genes that showed chromatin interactions in HiChIP and an overlap with 10 genes that were identified by CRISPRqtl to be regulated by these enhancers (Fig. 4G and Additional file 7: Table S6). Some of the inconsistencies between these datasets are likely due to the fact that for some enhancers the effect on gene expression in the context of this cell line is small or that the target gene is not detected at high enough levels in single cell RNASeq. Overall, these results show that regulation of gene expression through chromatin interactions is the most likely mechanism of action for these risk loci and demonstrate functional CRISPR screens as a highly reliable strategy for defining targets of GWAS hits.

## Discussion

GWAS have been highly successful in identifying variants associated with BC risk. Although a major goal of these studies is to identify new strategies for cancer prevention or treatment, a major obstacle in translating these findings to meaningful biological insights is that most risk variants are non-coding and the gene targets of the associations are not clear. Following fine mapping to identify the CCVs for BC, prioritizing loci with relatively few CCVs, chromatin conformation capture (3C) and luciferase assays have been performed at 16 BC risk loci implicating regulation of *TERT* [40], *CCND1* [41], *FGFR2* [42], *IGFBP5* [43], *MAP3K1* [44], *ESR1*, *RMND1* and *CCDC170* [45], *KLF4* [46], *NRBF2* [47], *ABHD8* [48], *FGF10* and *MRPS30* [49], *KLHDC7A*, *PIDD1*, *CITED4*, *PRKRIP1* and *RASA4* [4], *DUSP4* [27], *NTN4* [50], *TBX3* [35], and novel lncRNAs, *CUPID1*, and *CUPID2* [51].

Identifying GWAS gene targets and evaluating functional mechanisms at all known BC-risk loci individually is challenging. Expression quantitative trait locus (eQTL) links

a variant to a transcript and has been suggested as a powerful strategy to identify genes that are regulated by risk variants. Recent studies, however, demonstrate that for various reasons such as context dependent expression and sample size eQTL analysis is limited in its ability to identify target genes of risk variants [52]. Here, we show that functional CRISPR screens are a robust alternative for identification of target genes and the phenotypes they mediate. We used pooled CRISPR activation and suppression screens to simultaneously evaluate hundreds of putative GWAS target genes. This identified 20 genes predicted by INQUISIT with high-confidence to be GWAS targets, which mediate a cancer phenotype (Fig. 5). Although about half of these are already represented in a curated list of 278 established BC driver genes [53–57], the remainder were not previously implicated in BC biology. We identified an additional 46 genes that drive a proliferation or DNA damage response phenotype in breast cells, but most of these were predicted only with moderate confidence by INQUISIT to be target risk genes. Further studies will be needed to gain insights into their role in BC risk, and the mechanisms and pathways regulated by these genes. This proof-of-principle experiment demonstrates the utility of functional screens in identifying GWAS targets. Future studies using a similar approach with other cancer-related readouts will likely identify other GWAS hits that regulate different phenotypes.

| Risk region | Implicated Gene | Direction | Assay |
| --- | --- | --- | --- |
| chr2:24687599-25687599 | ADCY3 | Up | Proliferation |
| chr12:13913931-14913931 | ATF7IP | Down | Proliferation |
| chr11:107845515-108857137 | ATM | Down | DNA damage |
| chr3:63441697-64467900 | ATXN7 | Down | Proliferation |
| chr13:32468810-33472626 | BRCA2 | Down | DNA damage |
| chr1:10043820-11066215 | CASZ1 | Down | Proliferation |
| chr11:68831418-69879161 | CCND1 | Up | Proliferation |
| chr19:29777729-30777729 | CCNE1 | Up | Proliferation |
| chr11:65043827-66083066 | CFL1 | Down | Proliferation |
| chr11:65043827-66083066 | RNASEH2C | Down | DNA damage |
| chr16:3606788-4606788 | CREBBP | Down | Proliferation |
| chr8:29009616-30009616 | DUSP4 | Down | Proliferation |
| chr19:19048246-20048246 | LPAR2 | Down | Proliferation |
| chr19:19048246-20048246 | MAU2 | Down | DNA damage |
| chr8:127424659-130041931 | MYC | Up | Proliferation |
| chr17:28711667-29730520 | NF1 | Down | Proliferation |
| chr21:16073983-17073983 | NRIP1 | Down | Proliferation |
| chr21:16073983-17073983 | NRIP1 | Up | DNA damage |
| chr3:30175880-31175880 | TGFBR2 | Down | Proliferation |
| chr3:171785237-172785237 | TNFSF10 | Up | Proliferation |
| chr1:154648781-155648781 | RP11263K19.4 | Down | Proliferation |

**Fig. 5** Summary of genes scoring in CRISPR screens. Fine mapped risk regions that contain INQUISIT level 1 genes that scored in functional CRISPR screens

One of the strengths of our study was that we used four different phenotypic assays. Although most published CRISPR screens in breast cancer cell lines have used 2D culture, recent studies have shown the added utility of using 3D cell-based screens which more accurately measure cell proliferation than those carried out in 2D [58]. Similar to these studies, we also found that 3D proliferation assays gave a stronger and more robust signal. However, unlike genome wide 2D screens which failed to identify even known tumor-suppressor genes [58], here we show that in a smaller scale screen we are able to robustly identify known and new tumor-suppressor- and oncogenes (e.g., *MYC*, *CCND1*, *ATF7IP*, and *NRIP1*). This is likely due to the increased sensitivity that we achieved by increasing the number of cells infected with a given sgRNA (1000 cells/sgRNA as opposed to 300–500 cells/sgRNA typically used in whole genome screens [15]). Our results suggest that increasing the number of infected cells and sequencing reads increases sensitivity and enables robust detection of small proliferation changes. This should be considered in genome-wide gain of function CRISPR screens.

In few cases, we have identified a gene as both a tumor suppressor and an oncogene. This is consistent with previous observation showing that some genes could behave as a tumor suppressor or an oncogene depending on the context [59–66].

Using olaparib synthetic lethal screens, we identified BC risk genes that regulate the DNA repair pathway. Most of the hits we identified have been already identified in previous screens [33, 34], but six genes (including four INQUISIT level 1 genes, *CREBBP*, *NF1*, *MYC*, and *NRIP1*) have not. Interestingly, *NRIP1* showed variable phenotypes. In proliferation screens, suppression of *NRIP1* expression induced proliferation, and in DNA repair screens, over-expression of *NRIP1* showed an olaparib synthetic lethal phenotype. This may represent a larger set of genes that have a context-dependent phenotype.

Another strength of our study was that we used six immortal mammary cell lines, each with different characteristics. Because these lines are not transformed, they are ideal for identifying cancer initiation genes, which we expect to be the causal genes at many risk loci. Some genes scored in most cell lines, but the majority of genes only scored in a few. Although most genes did not pass our hit threshold in all cell lines, the phenotypes we observed were highly correlated between different cell lines (Additional file 1: Fig. S9D). However, some of the differences between cell lines following individual gene validation might be because the activity of the genes is context-specific. These observations demonstrate the robustness of the screens and show the importance of using multiple cell lines and multiple assays when measuring the effect of gene perturbation on phenotypes.

Enrichment analysis showed that INQUISIT level 1 genes were over-represented across all screen modalities, compared not only to background genes but also to INQUISIT level 2 genes, predicted with moderate-confidence, and genes identified by TWAS or eQTL analyses, thereby providing confidence in INQUISIT's ranking of putative target genes. Recently, several other algorithms that predict enhancer targets, including Activity By Contact (ABC), have been described [67, 68]. Of the functional genes detected in our screens, only 13 are predicted using the ABC method in breast derived samples. It is worth noting that, of the 21 INQUISIT level 1 target genes that scored in our CRISPR screens, eight are potentially impacted by CCVs through splicing or coding changes which are not considered by ABC.

INQUISIT did not identify any target genes for 15 of the 205 BC risk signals. For these, we therefore included all genes within a 2-Mb window centered on the risk signal (105 genes in total). Of these, only one gene (*JAZF1*) scored in our functional screens, and only with CRISPRi in two cell lines (hit rate of 1/105 < 1%), consistent with background detection levels. The CCVs at these loci may regulate genes in non-breast cell types, such as immune cells, or need specific stimuli; alternatively, the targets may be unannotated genes or non-coding RNAs. For example, we have recently identified several novel lncR-NAs, unannotated in public databases, which are regulated by BC risk variants [51, 69].

Using CRISPR activation and suppression screens, we found 13 genes, predicted by TWAS/eQTL analyses, which induced a proliferation or DNA damage phenotype. We did not validate or further pursue these 13 genes because we did not find any enrichment of hits among this class of genes, and others have shown inconsistencies in the direction of effect between TWAS findings and known Mendelian genes, including for BC [70]. However, some of them might be genuine BC risk genes.

The most common mechanism for non-coding risk variants is through chromatin interactions to regulate gene expression. Although most studies of enhancer function have utilized chromatin confirmation experiments, two major factors limit the utility of chromatin structure studies: (a) the majority of these studies are performed in cultured cells which may not recapitulate the native in vivo chromatin structure; (b) enhancers frequently interact with multiple genes making functional interpretation challenging. Here, we use a combination of functional CRISPR screens, HiChIP and CRISPRqtl, to identify chromatin interactions and phenotypes associated with BC-associated risk loci. Since the chromatin interaction experiments we performed were in cultured cell-lines, we cannot exclude the possibility that the chromatin interactions are different in vivo. In some cases, chromatin interactions are preserved in vitro and in vivo and we demonstrate the value of combining HiChIP and CRISPRqtl for identifying enhancer gene targets. HiChIP interacting regions can be large, and it can be difficult to pinpoint the exact region of association. In CRISPRqtl, the effective window is relatively small (up to 500 bp), and thus combining these approaches is likely to yield better resolution. In this study, we demonstrate the added value of functional phenotypic screens for identifying enhancer targets. Functional screens target the candidate genes rather than the CCV and thus a phenotype could be detected even if the chromatin interactions are not preserved in cultured cells or if the genes are impacted by coding or splicing variants.

## Conclusions

In summary, we demonstrate that pooled functional CRISPR screening is a cost-efficient, high-throughput, and robust method for identifying genes that are associated with BC risk loci. Application and extensions of this approach will be important for harnessing the benefits of cancer GWAS, and for translating genomic findings to treatments.

## Methods

### Cell lines

Human mammary epithelial cells (HMLE) [20] used in this study was a gift from Prof. William Hahn (Dana Farber Cancer Institute); the B80 cell lines (B80-T17 and B80-T5) are in vitro immortalized mammary cell lines previously described [18]. K5 + /K19 − and

K5+/K19+cell lines are immortalized progenitor mammary stem cells [17]. HMLE were induced to undergo epithelial to mesenchymal transition (EMT) to obtain a mesenchymal phenotype (mesHMLE) by culturing cells in DMEM to F12 media (1:1) supplemented with 10 μg/ml insulin, 20 ng/ml EGF, 0.5 μg/ml hydrocortisone, 5 μg/ml gentamycin, and 5% FBS treated with 2.5 ng/ml TGFβ1 for a minimum of 14 days [19]. HMLE and B80-T17 were propagated in mammary epithelial growth medium (MEGM) (Sigma). B80-T5 were cultured in RPMI 1640 (Sigma) supplemented with 10% FBS, 1% penicillin and streptomycin and 1% glutamine. K5+/K19−and K5+/K19−cells were maintained in DFCI medium containing: MEMα/Ham's F12 nutrient mixture (1:1, vol/vol) supplemented with 0.1 M HEPES, 1 μg/ml insulin, 1 μg/ml hydrocortisone, 12.5 ng/ml epidermal growth factor, 10 μg/ml transferrin, 14.1 μg/ml phosphoethanolamine, 0.545 ng/μl β-estradiol, 2 mM glutamine, 2.6 ng/ml sodium selenite, 1 ng/ml cholera toxin, 6.5 ng/ml triiodothyronine, 0.1 mM ethanolamine, 35 μg/ml bovine pituitary extract, 10 μg/ml gentamycin, and 10 μg/ml freshly prepared ascorbic acid. All cell lines were maintained in a humidified incubator at 37 °C with 5% $CO_2$. Cell line identity for established cells was confirmed by STR analysis at the Australian Genome Research Facility.

### Generation of stable cell lines

Lentiviral vector expressing a gene or sgRNA of interest, along with pMD2.G (Addgene#12,259) and psPAX2 (Addgene#12,260), was transfected into HEK293FT packaging cells (Thermo Fisher#R70007). Lentiviral supernatant was harvested after 48-h incubation in DMEM containing 30% FBS and passed through a 0.45-μm Milli-hex filter. For oncogenic potential: K5+/K19−and K5+/K19+cells were transduced with pLENTI-Hygro-PGK-TP53-DD and selected using 100 μg/ml hygromycin. For colony formation assays and in vivo assays: HMLE, mesHMLE, B80-T5, B80-T17, and K5+/K19+were transduced with pLX311-GFP-MEKDD and selected for GFP using the BD Influx™ cell sorter. K5+/K19−was transduced with pRRLsin-SV40 T antigen-IRES-mCherry (Addgene#58,993), and positive cells were sorted using the BD Influx™ cell sorter. For CRISPR screens and validations, all cell lines were transduced with following lentiviral vectors: Lenti-Cas9-2A-Blast (Addgene#73,310), Lenti-dCas9-KRAB-Blast (Addgene#89,567), and Lenti-dCas9-VP64-Blast (Addgene #61,425). Cells were selected and maintained in blasticidin (5 μg/ml to 10 μg/ml). For single gene perturbation, 3 sgRNAs were cloned into BsmBI-digested lenti-Guide-Puro vector (Addgene#52,963) for CRISPRko and pXPR502 vector (Addgene#96,923) for CRISPRa. Cells were infected with sgRNAs, selected, and maintained in puromycin (1 μg/ml to 2 μg/ml).

### RNA-seq

Transcriptome profiling was carried out using strand-specific TruSeq kit. Following RNA extraction (RNeasy, Qiagen) mRNA was enriched using polyT beads (Genewiz), and sequencing libraries were prepared using Illumina strand-specific TruSeq kit (Genewiz). Samples were sequenced on an Illumina HiSeq machine (PE 150 bp). RNA-seq were aligned to Ensembl v70 gene models with STAR v2.7.1a. Duplicate reads were marked with PicardTools v2.19, then reads mapping to transcriptome using featureCounts in subread v1.6.0, count matrix generated using RSEM v1.3.1. Differential

expression analysis was performed using DESeq2 in R v3.6.2. RNASeq data is available through GEO (GSE219168) [71].

### ATAC-seq

Profiling of regions of open chromatin using previously reported protocols [72]. Duplicate libraries were prepared for each cell type and paired-end sequenced (150 bp) generating a minimum of 40 filtered reads per library. Adapters trimmed using Cutadapt v1.13 and reads aligned to GRCh37 using Bowtie v2.2.9. Duplicates marked with Picard MarkDuplicates v2.19. Peaks were called using MACS2 and cell type-specific replicating peaks identified using BedTools. ATACSeq data is available through GEO (GSE219168) [71].

### HiChIP

HiChIP libraries were generated with the Arima HiChIP kit using an antibody against H3K27ac (Active Motif AbFlex: 91,193). Cells were counted using the Countess II automated cell counter (Thermo Scientific) and fixed with 2% formaldehyde using the Arima HiC + Kit (Arima, A101020). 1e6 fixed cells were used in restriction enzyme digest, biotin end filling ligation reactions to the manufacturer's protocol. Libraries were prepared using the KAPA Kit (KAPA, KK2620), according to the Arima-HiC kit protocol. Libraries were indexed using the Swift Biosciences indexing kit then paired-end sequenced (150 bp) with Illumina Novaseq 6000 to generate > 500 M raw reads per library. Individual replicate reads were processed with HiC-Pro (v 2.11.4) and aligned to hg19. Replicate samples for each cell type were quality controlled and checked for genome-wide signal correlation before merging with HiC-Pro. Enriched regions representing H3K27ac peaks were detected using MACS2. Chromatin loops were detected in each cell type-specific dataset using FitHiChIP v8.1 at 2 kb resolution limiting to 2 Mb interaction distance. Peak-to-peak, peak-to-nonpeak, and peak-to-all loops were used for background modeling and a $q < 0.01$ threshold set to determine significant interactions. HiChIP data is available through GEO (GSE219168) [71].

### Generation of pooled sgRNA library

sgRNA sequences in custom libraries are available in Additional file 3: Table S2. sgRNAs were designed using CRISPick algorithm [14]. For each gene, we chose top scoring 5 sgRNAs (based on CRISPick scores). Libraries were prepared as previously described [16, 22, 73]. Briefly, oligonucleotide pools (CustomArray) contained the sgRNA sequence appended to BsmBI cutting sites and overhang sequences for PCR amplification. The final sequence obtained is: AGGCACTTGCTCGTACGACGCGTCTCACACCG [20nt spacer]GTTTCGAGACGTTAAGGTGCCGGGCCCACAT. Following PCR amplification with Fwd: 5′-AGGCACTTGCTCGTACGACG-3′, Rev: 5′-ATGTGGGCCCGG CACCTTAA-3′ primers, the PCR product was cloned via Golden Gate assembly into BsmBI-digested lentiGuide-Puro vector (Addgene# 52,963) for CRISPRko and CRISPRi libraries and into pXRP502 (Addgene #96,923) for CRISPRqtl oligos were cloned into CROPseq-Guide-Puro (Addgene#86,708). Ligated libraries were electroporated into NEB5α electrocompetent cells (NEB), plasmid DNA was extracted using Qiagen Maxi Prep. For each library preparation, a 1000X representation was ensured.

### 2D and 3D proliferation screens

Mammary cell lines (HMLE, mesHMLE, B80-T5, B80-T17, K5 + /K19 − and K5 + / K19 + cells) stably expressing Cas9 (Addgene# 73,310), KRAB-dCas9 (Addgene# 89,567), or dCas9-VP64 (Addgene# 61,425) were established. Cells were then transduced with either the CRISPRko, CRISPRi, or CRISPRa Library at a MOI of 0.3 to obtain 1000 cells/sgRNA. Twenty-four hours post infection, cells were selected using puromycin (2 µg/ml) for 7 days. Cells were then subdivided to assay for 3D proliferation by plating cells in low attachment conditions (Corning#4615) or on 2D plates. To ensure sgRNA and Cas9/dCas9 expression, cells were maintained with puromycin and blasticidin throughout the screen. Twenty-one days post-infection, cells were washed in PBS, and genomic DNA was extracted using NucleoSpin Blood XL kit (Clontech). For colonies grown in low attachment conditions, genomic DNA was extracted using the DNeasy Kit (Qiagen). Libraries were prepared and sequenced as described below.

### Olaparib synthetic lethal screens

Olaparib synthetic lethal screens were done as previously described [33]. Cell lines stably expressing Cas9 (Addgene#73,310), KRAB-dCas9 (Addgene# 89,567), or dCas9-VP64 (Addgene#61,425) were transduced with the above described CRISPRko, CRISPRi, or CRISPRa sgRNA libraries (Additional file 3: Table S2) at a low MOI (0.3) at a coverage of 1000 cells/sgRNA. Puromycin containing medium was added 24 h post-infection, and cells were allowed to undergo selection for 7 days. For all screens, following selection, cells were trypsinized and divided into two treatment groups: DMSO or olaparib. HMLE, mesHMLE, B80-T17, K5 + /K19 − , and K5 + /K19 + cells were treated with 5 µM of olaparib, and B80-T5 cells were treated with 2.5 µM of olaparib for 14 days. Media was replaced every 4 days with DMSO or olaparib. Cells were harvested by centrifugation, and genomic DNA was extracted using NucleoSpin Blood XL kit (Clontech). Libraries were prepared and sequenced as described below.

### In vivo screen

HMLE-MEKDD, K5 + /K19 + -MEKDD, and B80-T5-MEKDD cells expressing Cas9 or dCas9-VP64 were infected at MOI = 0.3 with CRISPRko or CRISPRa libraries. Following puromycin selection (2 µg/ml) for 7 days, 2e6 cells/site were subcutaneously injected into NSG mice at 3 sites/mouse. Tumor growth was measured using a digital caliper every 48 h and monitored continuously until tumor volume reached 1 cm$^3$ (sum of all three sites). Tumor volume was calculated using the formula length (mm) × width (mm) × height (mm). Mice were sacrificed once tumors reached 1 cm$^3$. Cells were dissociated using Bead Ruptor machine and glass beads, and DNA was extracted using DNeasy Kit (Qiagen). Libraries were prepared and sequenced as described below.

### Library preparation, sequencing, and analysis

High-throughput sequencing library was generated using one-step PCR to amplify the integrated sequence within the construct and the addition of a barcode as previously described [16, 22, 73]. PCR products were then purified using AMPure beads

and samples sequenced using HiSeq (Illumina). PoolQ (https://portals.broadinstitute.org/gpp/public/software/poolq) was used for deconvolution and alignment of sgRNA reads.

### Crystal violet proliferation assay

Cells were plated at 2000 cells/well and allowed to propagate until confluent. Media was aspirated and washed twice in PBS followed by fixation in 10% formalin for 10 min at room temperature. Formalin was removed, and 0.5% (w/v) of crystal violet solution (Sigma) was added and incubated for 20 min at room temperature. Plates were washed in $dH_2O$ and imaged. For quantification, 10% acetic acid was added to each well and incubated at room temperature for 30 min. The crystal violet solution was quantified by measuring the OD at 590 nm using the PHERAstar (BMG).

### 3D proliferation assays

Cells were plated at 8000 cells/well in a 24-well low attachment plate (Corning). Colonies were allowed to form for 21 days. Images were taken at $\times 4$ magnification using an EVOS M5000 microscope (Thermo). Quantification of colonies was done by adding Cell-Titer-Glo Reagent (Promega) to wells, followed by a 10-min incubation at room temperature on a shaker. Cell lysates were transferred to a 96-well white plate and luminescence measured using the PHERAstar (BMG).

### sgRNAs for validation screens

All sgRNA sequences used in validation screens are in Additional file 8: Table S7.

### Western blot

Cells/tissue were harvested, washed in PBS, and resuspended in RIPA buffer (CST-9806) containing proteinase inhibitors (Roche) and quantified using the Pierce BCA Protein Assay Kit (Thermo Fisher). Protein lysates diluted in 4 X Laemmli Sample Buffer (Bio-Rad 161–0747) were loaded onto Bio-Rad 4–20% precast gels. Following electrophoresis, proteins were transferred to a pre-activated PVDF membrane using the Trans-Blot®Turbo™ Transfer System and visualized using ECL (Bio-Rad Chemidoc). Antibodies used in this study are DUSP4 (CST#5149), Cyclin D1 (CST#2978), Cyclin E1 CST#4129), ATF7IP (Sigma#16,578), ATF7IP (Sigma#HPA023505), ADCY3 (Abcam#ab199157), ATXN7 (Invitrogen#PAI-749), CREBBP (CST#7389), SAPK/JNK (CST#9252), LPAR2 (Abcam#ab135980), NF1 (Bethyl#A300-140A-M), Phospho-p38 MAPK (Thr180/Tyr182) (CST#4511), p38 MAPK (D13E1) (CST#8690), p44/42 MAPK (Erk1/2) (CST#4695), Phospho-p44/42 MAPK (Erk1/2) (Thr202/Tyr204) (CST#4370), Phospho-SAPK/JNK (Thr183/Tyr185) (CST#4668), TRAIL (CST#3219), RIP140 (Santa Cruz#sc518071), GAPDH (Santa Cruz#sc32233), c-Myc (CST#5605), TGFBR2 (Santa Cruz#sc17792), CASZ1 (Santa Cruz#sc398303), and CFL1 (Abcam#ab42824). Full blots are shown in Additional file 1: Fig. S11-S13.

### Animals

The Monash University Animal Ethics Committee approved all animal use in this study (AEC – approval number 2020–24,197-49,078). For these experiments, 5–7-week-old

female NSG mice were purchased from Australian Research Laboratories (WA, Australia) or were kindly gifted from Professor Gail Risbridger and A/Prof. Renea Taylor (Monash University).

### Validation of in vivo screens

B80-T5-MEKDD cells stably expressing Cas9 were infected with lentiviruses containing sgRNA's targeting *AAVS1* (control), *ATF7IP*, *DUSP4*, *TGFBR2*, and *CREBBP*. Twenty-four hours post-infection, cells underwent puromycin selection for 7 days and expanded. Cells were trypsinized, washed twice in PBS, and injected into NSG mice subcutaneously under isoflurane anesthesia. For each sgRNA, we injected 2e6 cells/site, 3 sites per mouse. Tumor growth was measured using a digital caliper every 48 h and monitored continuously until tumor volume reached 1 cm$^3$ (sum of all three sites). Tumor volume was calculated using the formula length (mm) × width (mm) × height (mm). Mice were sacrificed once tumors reached 1 cm$^3$.

### CRISPRqtl

CRISPRqtl was done as previously described [8]. Briefly, B80-T5 cells stably expressing KRAB-dCas9 were infected with the CRISPRqtl library at MOI = 5. Twenty-four hours post infection, cells were selected with puromycin (2 µg/ml) and cultured for 10 days. Cells were trypsinized washed with PBS and resuspended in PBS to reach a concentration of 1200 cells/ml. Single-cell suspensions were loaded on three lanes of a 10X Genomics Chromium Controller and Chromium Next GEM Single Cell 3′ GEM. We detected a total of 24,490 cells with a mean of 49,515 reads/cell and a median unique molecular identifier (UMI) of 15,186 UMIs/cell. Library and Gel Bead Kit v3.1 (10X Genomics cat #1,000,121), per manufacturer's instructions (CG000204 Rev D) with the following modifications and variables. A single sample was loaded in two wells of the Next Gem Chip G, overloaded at 150% of the recommended cell input volume, with the corresponding volume of dH2O deducted at Step 1.2b (using the Cell Suspension Volume Calculator Table; p26). At Step 2.2d, cDNA was generated using 11 cycles of PCR. Samples were recombined 1:1 before Step 3.1. Prior to enzymatic shearing, 10% of the cDNA was used for sgRNA PCR enrichment. Specifically, A three-step nested PCR was used for gRNA enrichment [38].

PCR 1: 5 ng of 10 × cDNA was amplified using NEBNext high fidelity 2 × PCR mix (NEB # M0541) and the following primers: Rxn1_Fwd: TTTCCCATGATTCCTTCA TATTTGC, Rxn1_Rev: ACACTCTTTCCCTACACGACG. Cycling conditions: 98 °C for 30 s, 14x (98 °C for 10 s, 50 °C for 10 s, 72 °C for 20 s), 72 °C for 2 min. PCR product was gel purified using the Qiagen MinElute Gel extraction kit (Qiagen # 28,604).

PCR 2: 5 ng of PCR 1 was amplified using NEBNext high fidelity 2 × PCR mix (NEB # M0541) and the following primers: Rnx2_Fwd: GTGACTGGAGTTCAGACGTGTGCT CTTCCGATCTTTGTGGAAAGGACGAAACAC, Rnx2_Rev: AATGATACGGCGACC ACCGAGATCTACACTCTTTCCCTACACGACGCTC. Cycling conditions: 98 °C for 30 s, 7x (98 °C for 10 s, 64 °C for 10 s, 72 °C for 15 s), 72 °C for 2 min. PCR product was gel purified using the Qiagen MinElute Gel extraction kit (Qiagene # 28,604).

PCR 3: 5 ng of PCR 1 was amplified using NEBNext high fidelity 2 × PCR mix (NEB # M0541) and the following primers: Rnx3_Fwd: CAAGCAGAAGACGGCATACGA

GATGACAGCATGTGACTGGAGTTCAGACGT, Rnx2_Rev (see PCR_2). Cycling conditions: 98 °C for 30 s, 11x (98 °C for 10 s, 64 °C for 10 s, 72 °C for 15 s), 72 °C for 2 min. PCR product was gel purified using the Qiagen MinElute Gel extraction kit (Qiagene # 28,604) and then purified using AMPure beads (Beckman Coulter # A63881). cDNA and PCR product were pooled at a 1:10 ration and sequenced on two lanes of an MGISeq machine (Genewiz) using 150 PE-cycles (total of 1212e6 reads).

CRISPRqtl was analyzed using the SCEPTRE algorithm as previously described [39].

## Supplementary Information

---

**Additional file 1: Fig. S1.** Example of genomic features used in INQUISIT to predict gene targets. **Fig. S2.** Characterization of immortalized cell lines used in this study. **Fig. S3.** Identification of genes that upon suppression or activation promote 2D or 3D growth. **Fig. S4.** Comparison between CRISPRko and CRISPRi screens. **Fig. S5.** Comparison between 2D and 3D screens. **Fig. S6.** Validation of sgRNAs used to suppress or activate INQUISIT level 1 hits. **Fig. S7.** Validation of INQUISIT level 1 hits that induce a 2D or 3D proliferation phenotype. **Fig. S8.** Identification of BC-risk genes that upon suppression or activation promote growth in immune deficient mice. **Fig. S9.** DUSP4 is a tumor-suppressor gene that is regulated by MEK1 expression and regulates phosphorylation of pJNK and pp38. **Fig. S10.** Identification of genes that upon suppression or activation modulate the DNA damage response. **Fig. S11.** Full Western blots of images shown in Fig. S6. **Fig. S12.** Full Western blots of images shown in Fig. 2. **Fig. S13.** Full Western blots of images shown in Fig. S9.

**Additional file 2: Table S1.** BC risk signals identified in GWAS and INQUSIT gene predictions. Signals from BC GWAS and INQUISIT gene prediction for these signals.

**Additional file 3: Table S2.** sgRNA sequences. Sequences of sgRNAs used in various CRISPR screens in this study.

**Additional file 4: Table S3.** Raw counts from different CRISPR screens. Raw sequencing reads from various CRISPR screens.

**Additional file 5: Table S4.** MAGeCK analysis from CRISPR screens. MAGeCK analysis for identification of enriched and depleted genes in CRISPR screens.

**Additional file 6: Table S5.** HiChIP interactions in K5+/K19+ and B80-T5 cells. Chromatin interactions between BC risk signals and genes at a 2Mb window.

**Additional file 7: Table S6.** Results from CRISPRqtl screen in B80-T5 cells. The Sceptre algorithm was used to identify gene expression changes following CRISPRi mediated targeting of 16 loci associated with increased BC risk.

**Additional file 8: Table S7.** Sequences of sgRNAs used in validation assays.

**Additional file 9.** Review history.

---

### Acknowledgements

We thank the Functional Genomics Platform, the Bioinformatics Platform, and Micromon genomics platform at Monash University for help with CRISPR screens, data analysis, and single cell experiments and Professor Gail Risbridger and A/Prof. Renea Taylor, Monash University, for providing NSG mice. We thank Dr. Timothy Barry and Dr. Eugene Katsevich for help with the SCEPTRE algorithm.

### Review history

The review history is available as Additional file 9.

### Peer review information

### Authors' contributions

Conceptualization, G.C.T and J.R; methodology, G.C.T., J.R., N.T., J.B., M.M., W.S., L.M., J.P., D.B., A.C., K.M. A.H., K.H., S.K., H.S., J.M.P, J.F., S.E. Analysis, J.B., J.R., N.T., D.P., L.J.P. Resources, R.R., V.M. Writing—original draft, J.R., G.C.T, J.B. Writing—review and editing, all authors; supervision, N.T., G.C.T, J.M.P, J.B. J.R. Funding acquisition, J.R. and G.C.T. The authors read and approved the final manuscript.

### Funding

This work was supported by a DoD grant to J.R. and G.C.T (grant number: W81XWH1910116). J.R is supported by a Victoria cancer agency fellowship (grant number: MCRF20035). G.C.T. is an NHMRC Leadership Fellow. S.L.E is an NHMRC Senior Research Fellow (grant number: 1135932). J.D.F. is supported by a philanthropic donation from Isabel and Roderic Allpass.

### Availability of data and materials

Full blots are shown in Additional file 1: Fig. S11-S13. All unique plasmids made in this study are available through Addgene. HiChip, RNA-Seq, and ATAC-Seq data generated in this study are available at GEO (GSE219168) [71]. All CRISPR functional screening raw and analyzed data is available in the supplementary Tables of this paper.

## Declarations

### Ethics approval and consent to participate

All animal studies were approved by the Monash University Animal Ethics Committee (AEC – approval number 2020–24197-49078).

### Competing interests

The authors declare that they have no competing interests.

### Author details

[1]Cancer Research Program and Department of Biochemistry and Molecular Biology, Biomedicine Discovery Institute, Monash University, Clayton, VIC, Australia. [2]Cancer Program, QIMR Berghofer Medical Research Institute, Brisbane, Australia. [3]Functional Genomics Platform, Monash University, Clayton, VIC, Australia. [4]Bioinformatics Platform, Monash University, Clayton, VIC, Australia. [5]Development and Stem Cells Program, Monash Biomedicine Discovery Institute, Clayton, VIC, Australia. [6]Cancer Research Unit, Children's Medical Research Institute, Faculty of Medicine and Health, The University of Sydney, Westmead, NSW, Australia. [7]Department of Genetics, Cell Biology and Anatomy, University of Nebraska Medical Center, Omaha, NE, USA.

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

## 