## [**Additional file 9.** Review history. · Genome Biology]

Review History

First round of review

Reviewer 1

Were you able to assess all statistics in the manuscript, including the appropriateness of statistical tests used? Yes, and I have assessed the statistics in my report.

Comments to author:

This study uses several functional genomics approaches (multiple CRISPR screens, single-cell CRISPR screens), performed in many different cellular models (2D, 3D, in vivo), to identify the gene targets of breast cancer risk loci. This study provides an incredibly number of resources to demonstrate the possibility to study the functions of potential genes that are involved in breast cancer proliferation and DNA damage. Of particular interest to the readers will be the combination of different screening technologies (CRISPR KO, CRISPRi, CRISPRa, single-cell) on multiple cell models. On the other hand, the organization and presentation of the data/results seem to be a major obstacle to clearly convey the messages in the manuscript.

Major concerns:

1. The presentation of the screening results are confusing to me (Fig. 1D, 2B, 3C). What does N=184(5%) mean? Also the relatively strong signals in "background" group in Figure 1 and Figure 3 imply that either these "background" genes are not really "backgrounds", or the selection threshold (FDR<.1, LFC>1 in 1 cell line) is not stringent enough. Maybe having a more stringent selection threshold will improve the interpretation.

2, Validation experiments (Fig S6/S7). The manuscript said "our approach identified 41 candidate BC risk genes" -- are they all validated? Fig S6/S7 seem to include only 14 genes. TGFBR2 and ADCY3 western blot results are missing. A summary table/figure of how many genes are tested and validated (in which aspect like western , 2D or 3D proliferation) would be useful for readers to understand the overall validation rate.

3, CRISPRqtl (single-cell CRISPR) is particularly exciting to me but looks like the results are not presented in a sufficient detail. For example, what does the clustering result (e.g., UMAP) look like? The authors mentioned some filtering procedure (>2 sgRNA UMI/cell). How many are filtered and how many are left? For the ATF7IP enhancer, how does the overall gene expression changes, including ATF7IP and nearby genes, upon perturbing this enhancer?

4, DNA damage screen (figure 3). The fact that only one hit shows up in K5+/K19+ row (last row) indicates the quality of that data may be poor, especially considering BRCA1 knockout didn't show up. Is that because K5+/K19+ already has mutant BRCA1/2? Also, this experiment requires filtering out hits that also show up in DMSO group. A description on how the filtering is done is missing.

Minor concerns:

1, the methods they used to analyze pooled screens and single-cell CRISPR screens (CRISPRqtl) are not clear. Looks like mageck is used for the analysis? Not clear about CRISPRqtl analysis? Please specify and cite relevant publications.

Reviewer 2

Were you able to assess all statistics in the manuscript, including the appropriateness of statistical tests used? Yes, and I have assessed the statistics in my report.

Comments to author:

The authors propose a CRISPR screen in 2D, 3D, and immune-deficient mice, to assess the downstream consequence of activation/suppression of GWAS target genes by modulating the promoters of these genes and observing a number of outcomes including proliferation, tumor formation, and DNA damage response. The experiments, data, and analyses would be a valuable contribution to the field of understanding GWAS loci via functional perturbation assays. In particular the multiple considerations for phenotype, and direct perturbation of candidate BC GWAS genes is especially useful. However, a few of the analyses could use more description and clearer figures, and the enrichment analyses are currently not well motivated.

Major comments

- The manuscript has: "Five positive controls showed dramatically increased sgRNA abundance and ten INQUISIT-predicted genes (five at Level 1) scored in this assay, suggesting these are potent drivers of BC-risk (Fig. 2B)", but it wasn't clear from Fig 2B how the positive controls or INQUISIT genes scored higher than negative controls (what was "background" here?), or how to interpret this major result. Figure 2B did not clearly distinguish the results of this experiment. What are the numbers and percents above each section of squares? How to interpret "N experiments"? The caption also doesn't provide much more detail: "(A) Experimental approach. (B) Summary of hits from the in vivo screens".

- "We tested for the enrichment of genes predicted by various computational and statistical methods amongst the combined hits from all phenotypic screens": how is it possible to test for enrichment, given that the experimental design is focused primarily on genes predicted by INQUISIT? What is being compared to? The genes in the 2Mb regions? In this case, the details of how those were selected will have a large impact on the enrichment analysis.

Again in the Discussion: "Enrichment analysis showed that INQUISIT Level 1 genes were over-represented across all screen modalities, compared not only to background genes but also to INQUISIT Level 2 genes, predicted with moderate-confidence, and genes identified by TWAS or eQTL analyses, thereby providing confidence in INQUISIT's ranking of putative target genes": This depends on how background genes were chosen. A naive background would make even a simple ranking strategy provide significant enrichment. For example, selecting background genes regardless of their expression level (thereby including many unexpressed genes in background) would provide statistical enrichment in a perturbation screen for many various algorithms for gene prioritization.

- "For 18 of the 21 INQUISIT Level 1 hits identified in the screens we found chromatin interactions with regions containing BC risk variants": is there a negative control for this experiment?

Minor comments

- It could be stated more clearly that the modulation of the GWAS target genes is at the promoter. This could be inferred from the Introduction and Methods, but it wasn't stated explicitly.
- For gene selection, the authors mention: "genes located within 2Mb of 15 risk signals...", was this all the genes within 2Mb? If not how were genes selected?
- "We then intersected these regions with BC risk CCVs": was this just using the lead GWAS variant, and asking if it falls within the 1kb region?
- It would help to briefly define the variables and weighting scheme used in INQUISIT, to better interpret the results on this experiment.

Detailed response to reviewers' comments:

Reviewer #1: This study uses several functional genomics approaches (multiple CRISPR screens, single-cell CRISPR screens), performed in many different cellular models (2D, 3D, in vivo), to identify the gene targets of breast cancer risk loci. This study provides an incredibly number of resources to demonstrate the possibility to study the functions of potential genes that are involved in breast cancer proliferation and DNA damage. Of particular interest to the readers will be the combination of different screening technologies (CRISPR KO, CRISPRi, CRISPRa, single-cell) on multiple cell models. On the other hand, the organization and presentation of the data/results seem to be a major obstacle to clearly convey the messages in the manuscript.

Major concerns:

1. The presentation of the screening results are confusing to me (Fig. 1D, 2B, 3C). What does N=184(5%) mean? Also the relatively strong signals in "background" group in Figure 1 and Figure 3 imply that either these "background" genes are not really "backgrounds", or the selection threshold ($FDR < .1$, $LFC > 1$ in 1 cell line) is not stringent enough. Maybe having a more stringent selection threshold will improve the interpretation.

We are sorry for not presenting clearly enough. 'N' represents the number of genes in a class and (X%) represents the percent of scoring genes. To make this clearer we modified Figures 1,2 and 3. In the revised manuscript we added to the captions a better description of these annotations.

Background genes are genes that scored with low confidence in INQUISIT and are not likely to be targets of risk loci. These are important controls to assess the reliability of these methods in predicting GWAS targets. Since many genes in the human genome will have an effect on fundamental phenotypes such as proliferation, to identify if a prediction method is actually enriched for real targets, it is essential to have a control. For example, *FLI1* is a known tumor suppressor and was included as a background gene. As expected, suppression of *FLI1* induced proliferation. Our results show that from a random list of genes ~3% score as -proliferation genes, which is important for our ability to assess which of the gene prediction methods we used is enriched with targets we could experimentally validate. From our results, it is very clear that TWAS or INQUISIT 2 genes are similar to background but that INQUISIT 1 genes are enriched with targets that we could functionally validate. To clarify this point, and address the reviewer's comment, we added more details of the background gene in the discussion (page 3 and 6 of the revised manuscript).

2. *Validation experiments (Fig S6/S7). The manuscript said "our approach identified 41 candidate BC risk genes" -- are they all validated? Fig S6/S7 seem to include only 14 genes. TGFBR2 and ADCY3 western blot results are missing. A summary table/figure of how many genes are tested and validated (in which aspect like western, 2D or 3D proliferation) would be useful for readers to understand the overall validation rate.*

We are sorry for the confusion. Our screens (proliferation in 2D, 3D and in vivo and DNA damage screens) identified a total of 66 genes that are predicted by INQ_1, TWAS or INQ_2 to be associated with a BC risk loci. Of these we validated the phenotypes of all INQUISIT 1 genes that scored in proliferation screens with CRISPRa or CRISPRko (14 genes) and 4 genes that scored in the DNA damage screen. We added a summary sentence to better describe these results (page 4 of the revised manuscript).

As requested by the reviewer we added to the revised manuscript a western blot showing TGFBR2 knockout. Unfortunately, we could not find any good antibodies for detection of ADCY3, so for ADCY3 we used qRT-PCR to validate gene expression changes induced by sgRNAs targeting *ADCY3* (Supplementary Fig. S6).

3. *CRISPRqtl (single-cell CRISPR) is particularly exciting to me but looks like the results are not presented in a sufficient detail. For example, what does the clustering result (e.g., UMAP) look like? The authors mentioned some filtering procedure (>2 sgRNA UMI/cell). How many are filtered and how many are left? For the ATF7IP enhancer, how does the overall gene expression changes, including ATF7IP and nearby genes, upon perturbing this enhancer?*

The CRISPRqtl experiment we did in this manuscript is similar to previously published CRISPRqtl analyses (Gasparini et al. Cell 2019) and we used the SCEPTRE pipeline for the analysis (Barry et al. Genome Biology 2021). In this experiment, cells are infected at high MOI so each cell will have on average 5 sgRNAs. To identify expression changes induced by an enhancer targeting sgRNA an average of all cells containing a sgRNA is used. This approach is very effective at identifying defined gene expression changes induced by enhancers. However, because every cell contains multiple sgRNAs and the overall expression profile is a mix of several sgRNAs targeting different enhancers it is not possible to cluster individual cells based on their gene expression profiles.

Filtering is done in this type of experiment to avoid non-specific noise. We typically see a low threshold background of sgRNA expression which is likely due to cell lysis (see Hill et al. Nature Methods, 2017). In addition, we filtered out cells with high mitochondrial content and cells with < 200 genes detected. This resulted in 13,667 cells. A sentence describing this in more detail was added to the revised manuscript (page 8 of the revised manuscript).

All changes that we measured in this experiment, including for *ATF7IP*, are in Supplementary Table S6 (SCEPTRE score tab). In the case of *ATF7IP*, CRISPRqtl found 6 genes that are regulated by the risk associated enhancer (*ATF7IP*, *MANSC1*, *RPL13AP20*, *EPS8*, *APOLD1*, *HIST4H4*). Of these genes only *ATF7IP* scored in our functional screens, thereby demonstrating the power of combining phenotypic screens with other data types.

4. DNA damage screen (figure 3). The fact that only one hit shows up in K5+/K19+ row (last row) indicates the quality of that data may be poor, especially considering BRCA1 knockout didn't show up. Is that because K5+/K19+ already has mutant BRCA1/2? Also, this experiment requires filtering out hits that also show up in DMSO group. A description on how the filtering is done is missing.

All cell lines used in this screen are normal immortalized breast cells with wild type *BRCA1/2*. We agree with the reviewer that for reasons that we do not understand *K5+/K19+* performed very poorly in the Olaparib synthetic lethality screen. The only gene that scored as a hit in this cell line in the DNA damage screen is *RANBP1* and no positive controls scored. We have decided to remove this cell line from the DNA damage screen in the revised manuscript, and *RNABP1* from the total number of hits we identified in this screen.

The filtering of hits that also show up in the DMSO arm is done in the analysis. Hits are identified by comparing sgRNA reads in DMSO and olaparib treated cells so genes that score in both DMSO and olaparib won't show up. We added a sentence better describing this in the revised manuscript (page 6 of the revised manuscript).

Minor concerns:

1. the methods they used to analyze pooled screens and single-cell CRISPR screens (CRISPRqtl) are not clear. Looks like mageck is used for the analysis? Not clear about CRISPRqtl analysis? Please specify and cite relevant publications.

In CRISPRqtl we used the previously published SCEPTRE analytical pipeline (Barry et al. Genome Biology 2021). We are sorry for not providing a clear enough description and have added more details in the method section of the revised manuscript describing CRISPRqtl and SCEPTRE analysis. We have now added an additional reference to the paper describing SCEPTRE (pages 8 and 16 of the revised manuscript).

Reviewer #2: The authors propose a CRISPR screen in 2D, 3D, and immune-deficient mice, to assess the downstream consequence of activation/suppression of GWAS target genes by modulating the promoters of these genes and observing a number of outcomes including proliferation, tumor formation, and DNA damage response. The

experiments, data, and analyses would be a valuable contribution to the field of understanding GWAS loci via functional perturbation assays. In particular the multiple considerations for phenotype, and direct perturbation of candidate BC GWAS genes is especially useful. However, a few of the analyses could use more description and clearer figures, and the enrichment analyses are currently not well motivated.

Major comments

- The manuscript has: "Five positive controls showed dramatically increased sgRNA abundance and ten INQUISIT-predicted genes (five at Level 1) scored in this assay, suggesting these are potent drivers of BC-risk (Fig. 2B)", but it wasn't clear from Fig 2B how the positive controls or INQUISIT genes scored higher than negative controls (what was "background" here?), or how to interpret this major result. Figure 2B did not clearly distinguish the results of this experiment. What are the numbers and percents above each section of squares? How to interpret "N experiments"? The caption also doesn't provide much more detail: "(A) Experimental approach. (B) Summary of hits from the in vivo screens".

We are sorry for not being clearer about this. In the results described in this manuscript we used well established and very commonly used methods to calculate sgRNA abundance. This is done using the MAGeCK algorithm (Li et al, Genome Biology, 2014). In this analysis the $\text{Log}_2[\text{Fold change}]$ in sgRNA abundance is calculated by comparing sgRNA reads in the sample to the DNA pool. The negative controls are sgRNAs that target non-human genes or AAVS1. In Fig. 2B only genes that scored as hits are presented. In the revised manuscript, we added a sentence to make this point clearer (page 3 of the revised manuscript).

"N experiments" refers to the number of experiments (assays) that a gene scores in. To make this clearer we modified the captions for Fig. 1-3 with better annotations.

- "We tested for the enrichment of genes predicted by various computational and statistical methods amongst the combined hits from all phenotypic screens": how is it possible to test for enrichment, given that the experimental design is focused primarily on genes predicted by INQUISIT? What is being compared to? The genes in the 2Mb regions? In this case, the details of how those were selected will have a large impact on the enrichment analysis.

We are sorry this was unclear. We used several methods to predict target genes at GWAS loci. The main method to select candidate genes was INQUISIT. This method assigns a score to reflect confidence for each predicted gene: the best supported targets are designated as "Level 1" and the least supported as "Level 3". We consider the Level 1 genes as the most likely targets, and therefore we used the enrichment analysis to test for the representation of Level 1 genes amongst the hits we detected across the screens. The comparison was therefore the fraction of hits which

correspond to Level 1 genes, divided by the fraction of Level 1 genes in the library. We have reworded the introductory sentence for this section on page (page 6 of the revised manuscript) accordingly.

"We tested for the enrichment of genes predicted by various computational and statistical methods amongst the combined hits from all phenotypic screens": how is it possible to test for enrichment, given that the experimental design is focused primarily on genes predicted by INQUISIT? What is being compared to? The genes in the 2Mb regions? In this case, the details of how those were selected will have a large impact on the enrichment analysis.

Again in the Discussion: "Enrichment analysis showed that INQUISIT Level 1 genes were over-represented across all screen modalities, compared not only to background genes but also to INQUISIT Level 2 genes, predicted with moderate-confidence, and genes identified by TWAS or eQTL analyses, thereby providing confidence in INQUISIT's ranking of putative target genes": This depends on how background genes were chosen. A naive background would make even a simple ranking strategy provide significant enrichment. For example, selecting background genes regardless of their expression level (thereby including many unexpressed genes in background) would provide statistical enrichment in a perturbation screen for many various algorithms for gene prioritization.

We thank the reviewer for this comment. We compared the genes selected by a specific method to the remaining genes in the library. The background can be considered as genes which had the potential to affect a phenotype, but which fell below the cutoff to be classified as a hit.

- "For 18 of the 21 INQUISIT Level 1 hits identified in the screens we found chromatin interactions with regions containing BC risk variants": is there a negative control for this experiment

We are not sure what the reviewer is referring to as a negative control. It is possible to compute the expectation under the null model, assuming that chromatin interactions connect breast cancer associated risk SNPs to genes at random. At the loci harboring the hits from our screen, we find 2,984 significant chromatin interactions which contain a risk SNP. Of these, 600 interactions link a risk SNP to a gene, of which 185 interactions link to one of the 18 genes detected in the phenotypic screens (removing *BRCA2* as the functional impact of the GWAS signal is most likely through a nonsense mutation; *NF1* and *RP11-263K19.4* were not detected by HiChIP). By repeated sampling of sets of 600 random interactions from the pool of 2984 (10,000 times), we see that the number of interactions between risk SNP and a screen hit is on average 53.48 (sd = 6.27). The observed number of significant interactions to functional genes (185) is therefore 3.5-fold greater than this random expectation. This is explained in page 7 of the revised manuscript.

Minor comments

- *It could be stated more clearly that the modulation of the GWAS target genes is at the promoter. This could be inferred from the Introduction and Methods, but it wasn't stated explicitly.*

To address the reviewer's comment, we modified the text to emphasize that in CRISPRi and CRISPRa screens the promoter is targeted (page 3 of the revised manuscript).

- *For gene selection, the authors mention: "genes located within 2Mb of 15 risk signals...", was this all the genes within 2Mb? If not how were genes selected?*

Yes, in this case all the genes in those regions were selected (no specific criteria were used). All the genes and sgRNAs used in these screens are described in detail in Supplementary Table S2.

- *"We then intersected these regions with BC risk CCVs": was this just using the lead GWAS variant, and asking if it falls within the 1kb region?*

Yes, that is correct. We modified the text to better describe this (page 7 in the revised manuscript).

- *It would help to briefly define the variables and weighting scheme used in INQUISIT, to better interpret the results on this experiment.*

INQUISIT has been extensively described in two publications (Fachal et al., 2020; Michailidou et al., 2017), and we added an example of INQUISIT in Supplementary Fig. S1, which is why we did not add all the details of INQUISIT to the manuscript, but we can if the reviewers would prefer. It was designed to rank the predicted target genes at BC risk loci. We used *in silico* data from breast tissue and cell lines to determine whether CCVs are likely to act via distal gene regulation, proximal gene regulation, or by impacting the gene's protein product. INQUISIT treats any CCV as potentially able to regulate distal genes and awards points to each gene based on: 1) chromatin interaction data from Capture-Hi-C and ChIA-PET experiments; 2) enhancer annotations based on computational methods designed to infer target genes from genomic data; 3) expression quantitative trait loci (eQTL) analysis of genes within 2 Mb of either side of each CCV when the risk and eQTL signals co-localize; 4) integration of transcription factor ChIP-seq data for specific proteins in breast cells shown to be positive predictors of BC CCVs. The intersection of CCVs, enhancers and these transcription factor binding sites resulted in up-weighting of the associated gene (see example of target gene rankings at a BC risk locus in Supplementary Fig. 1). Promoter variants were assessed for overlap with chromatin signatures characteristic

of transcription start sites (TSS) in breast cell lines and primary tissue, as well as putative functional transcription factor binding sites, gene expression data and eQTLs. Intragenic variants were evaluated for consequences of coding and splicing changes. We designated INQUISIT predictions with the strongest supporting evidence as Level 1, and Level 3 the lowest.

Second round of review

Reviewer 1: Most of my previous concerns have been properly addressed. One remaining issue is the CRISPRqtl analysis. First, in terms of the clustering analysis, the authors argued "because every cell contains multiple sgRNAs and the overall expression profile is a mix of several sgRNAs targeting different enhancers it is not possible to cluster individual cells based on their gene expression profiles.". This is definitely not the case -- the gene expression analysis (tSNE or UMAP) does not rely on the assignment of sgRNAs. It may be true that the multiple sgRNAs per cell may make it difficult to tell whether the expression changes of single cells are coming from perturbing a particular enhancers. However that should not affect the unbiased, expression-based analysis.

Second, the CRISPRqtl results are very useful to reveal the the causal effect of perturbing an enhancers that change target gene expression, however the current presentation (Figure 4) did not provide enough details. For example, Fig. 4g only shows the overlap between CRISPR screen and CRISPRqtl. For the bold genes (which are considered as the true target gene), does CRISPRqtl show stronger signals than nearby genes (i.e., genes that are not bold in Fig. 4g)? If you look at the target gene expression (e.g., MYC), do single cells containing sgRNAs targeting the corresponding enhancer have a lower MYC expression than other single cells? The same analysis is recommended in Fig. 4H-I to show single cells with sgRNAs perturbing ATF7IP enhancers did have lower ATF7IP expression.

Reviewer 2: While some of the explanations in the response were useful, as both reviewers were confused by the terms "background" and the diagrams 1D, 2B, 3C, the authors haven't done much to alleviate this confusion in the revised manuscript. The captions still don't guide the reader to understanding how to interpret these, e.g. "Summary of results", "Summary of hits", "Summary of hits". In the Figure "Number Assays" could be better described, it's the sum across the rows below, and doesn't take into account conflicting evidence at all. In my understanding -- again I'm just guessing -- the color (red/green) is reflecting the authors expectation given the experimental design, which is why a gene can be both labelled as tumor suppressor and oncogene, e.g. TNFSF10 is both. Given that two reviewers brought this up, it would be worth considering the readers and clarifying these plots in the caption, as this is arguably one of the major results of the paper. Note that the label "CONTROL" is not obviously interpreted. A close reader may notice that 16 is the same number as in: "16 known tumor-suppressor genes and oncogenes".

Finally, the authors should present the baseline expression level of the genes across the different sets of genes. If background genes are lowly expressed as I mentioned in my first review, this could explain a portion of the decreased rate of hits. The response to my initial comment didn't make this point clear: "The background can be considered as genes which had the potential to affect a phenotype, but which fell below the cutoff to be classified as a hit".

30/12/2023

Dear Dr. Bishop,

Thank you for the insightful reviews and the opportunity to revise our manuscript entitled 'CRISPR screens identify gene targets at breast cancer risk loci'. As you can see in the revised manuscript, we have addressed all the reviewers' comments and have added some clarifications (see below our detailed point-by-point response).

As requested, all the datasets in this manuscript have now been deposited. All the data for CRISPR screens and CRISPRqtl screens are available as Supplementary Tables. The MEK-DD vector has been deposited at Addgene (#81967) and will be made publicly available. RNASeq ,ATACSeq and HiSeq data have been deposited in GEO (GSE219168). We added a statement regarding data available in the methods section.

We hope that you will find our revised manuscript suitable for publication in Genome Biology.

Sincerely,
Joseph Rosenbluh, PhD
Monash University,
Australia

Georgia Chenevix-Trench, PhD
QIMR-Berghofer,
Australia

Detailed response to reviewers' comments:

Reviewer #1: Most of my previous concerns have been properly addressed. One remaining issue is the CRISPRqtl analysis. First, in terms of the clustering analysis, the authors argued "because every cell contains multiple sgRNAs and the overall expression profile is a mix of several sgRNAs targeting different enhancers it is not possible to cluster individual cells based on their gene expression profiles.". This is definitely not the case -- the gene expression analysis (tSNE or UMAP) does not rely on the assignment of sgRNAs. It may be true that the multiple sgRNAs per cell may make it difficult to tell whether the expression changes of single cells are coming from perturbing a particular enhancers. However that should not affect the unbiased, expression-based analysis.

We agree with the reviewer that it is possible to do a cluster analysis on these cells (see Reviewer Figure 1A). However, this is not an informative analysis because each cell has multiple sgRNAs. To better explain this, we have done a UMAP cluster analysis which shows 13 different clusters (Reviewer Figure 1A). Overlapping the sgRNA identity for a particular enhancer (see examples for *ATF7IP*, *MYC* and *NRIP1* enhancers (Reviewer Figure 1B-D)) shows that the sgRNAs are not located in a particular cluster. This is because each individual cell has multiple sgRNAs and thus many gene expression changes some related to a particular enhancer targeting sgRNA and some related to other enhancer targeting sgRNAs.

Reviewer Figure 1: UMAP clustering of CRISPRqtl experiment. (A) UMAP clustering of single cells based on CRISPRqtl experiment. (B) Overlay of all sgRNAs targeting fine mapped enhancer at chr12:13913931-14913931 (enhancer containing *ATF7IP*). (C) Overlay of all sgRNAs targeting fine mapped enhancer at chr8:127424659-130041931 (enhancer containing *MYC*). (D) Overlay of all sgRNAs targeting fine mapped enhancer at chr21:16073983-17073983 (enhancer containing *NRIP1*).

Second, the CRISPRqtl results are very useful to reveal the causal effect of perturbing an enhancers that change target gene expression, however the current presentation (Figure 4) did not provide enough details. For example, Fig. 4g only shows the overlap between CRISPR screen and CRISPRqtl. For the bold genes (which are considered as the true target gene), does CRISPRqtl show stronger signals than nearby genes (i.e., genes that are not bold in Fig. 4g)? If you look at the target gene expression (e.g., MYC), do single cells containing sgRNAs targeting the corresponding enhancer have a lower MYC expression than other single cells? The same analysis is recommended in Fig. 4H-I to show single cells with sgRNAs perturbing ATF7IP enhancers did have lower ATF7IP expression.

All the CRISPRqtl data and analysis are available in the supplementary tables of this manuscript. Supplementary Table S6 has all the SCEPTRE scores for every sgRNA used in CRISPRqtl. Specifically, in the third tab of this table (labelled: "Enh-Gene-sgRNA-summ-table") are the actual numbers for each enhancer-gene pair.

The reviewer asked whether the target gene showed the strongest signal in CRISPRqtl. This will depend on the peak that is used. For example, see below the CRISPRqtl data (data available in Supplementary Table S6) for all the enhancer peaks, that we used, around *MYC* (the same is true for the other enhancers we tested). Of the 14 enhancer peaks we tested, *MYC* scored as a hit in 6 enhancers (Sceptre score < -1). In 4 of those *MYC* was the strongest signal but in two peaks other genes (e.g. *FAM84B*, *ASAP1*, *PVT1*) had stronger signals (Reviewer Fig. 2A). Similarly, *ATF7IP* scored as a hit (Sceptre score < -1) in 6 of the 9 enhancer peaks. In 5 of those peaks *ATF7IP* was the strongest hit but in one peak *MANSC1* was stronger (Reviewer Fig. 2B).

It is expected that multiple genes score in CRISPRqtl because enhancers regulate the expression of more than one gene. In fact, this is one of the main reasons we used a functional readout in addition to defining gene expression changes. Our strategy of combining functional readouts (CRISPR screens) with genomic readouts (HiChIP and CRISPRqtl) is very powerful and enables pinpointing the gene targets that mediate breast cancer risk.

Reviewer Figure 2: CRISPRqtl hits in two fine mapped GWAS loci. For each gene enhancer pair the Sceptre score is plotted. The gene that also scored in CRISPR functional screens is marked in red. (A) CRISPRqtl of genes in enhancer peaks of chr8:127424659-130041931 (B) CRISPRqtl of genes in enhancer peaks of chr12:13913931-14913931

Reviewer #2: While some of the explanations in the response were useful, as both reviewers were confused by the terms "background" and the diagrams 1D, 2B, 3C, the authors haven't done much to alleviate this confusion in the revised manuscript. The captions still don't guide the reader to understanding how to interpret these, e.g. "Summary of results", "Summary of hits", "Summary of hits".

To better address the reviewer's concern, we modified the figure legend. To more accurately describe the background genes we modified the caption and now instead of "BACKGROUND" we labelled these genes as "GENES WITH NO SUPPORTING FUNCTIONAL EVIDENCE"

In the Figure "Number Assays" could be better described, it's the sum across the rows below, and doesn't take into account conflicting evidence at all.

We changed the figure legend to "Sum of assays in which a gene scored as a hit"

In my understanding -- again I'm just guessing -- the color (red/green) is reflecting the authors expectation given the experimental design, which is why a gene can be both labelled as tumor suppressor and oncogene, e.g. TNFSF10 is both.

As described in the manuscript (page 4, first paragraph) a tumor suppressor is a gene that upon suppression induces a cancer phenotype and an oncogene is a gene that upon overexpression induces a cancer phenotype. It is well documented that some genes could function as both a tumor suppressor and oncogene (e.g. TP53) and this is what we have observed for a small number of genes. This phenomenon is well documented (see for example PMID: 33115801, 32587399, 32547225, 29735662, 31782549, 32194886, 31724223, 16247450). We have described this in the manuscript (page 4, 3'd paragraph) but to make this point clearer we have added these references and an additional paragraph that explains this point in the discussion of the revised manuscript (page 9 of the revised manuscript).

Given that two reviewers brought this up, it would be worth considering the readers and clarifying these plots in the caption, as this is arguably one of the major results of the paper. Note that the label "CONTROL" is not obviously interpreted. A close reader may notice that 16 is the same number as in: "16 known tumor-suppressor genes and oncogenes".

To better address this we modified the figure caption. Instead of CONTROL we labelled these as "KNOWN ONCOGENES/TSG".

Finally, the authors should present the baseline expression level of the genes across the different sets of genes. If background genes are lowly expressed as I mentioned in my first review, this could explain a portion of the decreased rate of hits. The response to my initial comment didn't make this point clear: "The background can be considered as genes which had the potential to affect a phenotype, but which fell below the cutoff to be classified as a hit".

To address the reviewer's comment, we looked at gene expression levels across the different gene sets (see reviewer figure 3). As suggested by the reviewer, background genes do show a lower level of overall expression. However, this is because one of the criteria for selecting genes as hits in INQUISIT (and other prediction algorithms) is that they are expressed at medium to high levels. Thus, it is not surprising that background genes have the lowest levels of expression. Despite this, the proportion of scoring genes with no supporting evidence, and in the INQ_2 or TWAS selected genes, are very similar, even though INQ_1, INQ_2 and TWAS genes have similar levels of gene expression. This demonstrates that the low number of genes that score in the genes with no supporting evidence is not due to the lower expression levels. To better explain this point, we added these new supplementary figures (Supplementary figures S2D,E in the revised manuscript) and new text describing these observations (page 3 and 6 of the revised manuscript).

Reviewer Figure 3: Expression of genes in various groups. (A) Average gene expression across 6 cell lines that were used in this study for each category of genes. (B) Gene expression for each category of genes for each one of the cell lines used in this study.

Third round of review

Reviewer 1:

Authors did not address one question in my 2nd concern. My question is "do single cells containing sgRNAs targeting the corresponding enhancer have a lower MYC expression than other single cells?". Unfortunately this question was not answered. An enhancer is defined as a non-coding element that modulates expression. Therefore, perturbing that enhancer should see a reduction of target gene expression. That is the direct evidence of the enhancer-gene pair. The Sceptre score analysis (Reviewer Figure 2) is good but did not answer this question.

Reviewer 2:

The authors have responded to my previous comments in the revision.

Reviewer #1 comment: Authors did not address one question in my 2nd concern. My question is "do single cells containing sgRNAs targeting the corresponding enhancer have a lower *MYC* expression than other single cells?". Unfortunately this question was not answered. An enhancer is defined as a non-coding element that modulates expression. Therefore, perturbing that enhancer should see a reduction of target gene expression. That is the direct evidence of the enhancer-gene pair. The Sceptre score analysis (Reviewer Figure 2) is good but did not answer this question.

Response to reviewer 1:

The reviewer is asking if sgRNAs targeting the enhancer result in lower expression of the transcript. This is exactly what SCEPTRE does and what the experiment we did measured. The SCEPTRE score calculates the difference in gene expression between cells expressing an enhancer targeting sgRNA and all other cells. The SCEPTRE algorithm uses an average of all the sgRNAs targeting a particular enhancer. In our experiment we designed sgRNAs in intervals of 100bp within the enhancer (all sgRNA sequences used in CRISPRqtl are listed in Table S2).

If we understand the reviewer’s question, he/she is asking about single sgRNAs and not the average of multiple enhancer targeting sgRNAs (which is what is calculated in SCEPTRE). We and others using CRISPRqtl (for example PMIDs: 32483332, 30849375) typically don’t use individual sgRNAs and prefer to use methods such as SCEPTRE which combines scores from multiple sgRNAs and looks at the distribution of enhancer targeting sgRNAs. Looking at each sgRNA individually could be confusing and misleading.

However, to respond to the reviewer’s comment, we added below a plot showing all the sgRNAs targeting one of the *MYC* enhancers. Figure 1A shows the SCEPTRE scores for one of the *MYC*-targeting enhancers. This analysis uses the average of the 17 sgRNAs targeting this region and compares the distribution of these sgRNAs to distribution of cells without any *MYC* targeting sgRNAs. Fig. 1B shows the results for the individual sgRNAs. To avoid confounding factors, we removed all cells that contained more than one *MYC*-targeting sgRNA (these are not removed in the SCEPTRE analysis). As is evident from this plot, most sgRNAs targeting this enhancer show good reduction in *MYC* expression. However, from these results alone it is difficult to determine the significance of the results which is why we don’t use these results but instead use SCEPTRE.

Figure 1: SCEPTRE and individual sgRNA results from CRISPRqtl at the enhancer in chr8:129189076-129190790. (A) SCEPTRE analysis of genes located 2MB from the enhancer. (B) Expression levels in individual cells containing no *MYC* targeting sgRNAs or cells containing the indicated *MYC* targeting sgRNAs. Cells with more than one *MYC* targeting sgRNA were eliminated

Fourth round of review

Reviewer 1: The reason that I request examining target expressions between cells with (or without) an enhancer-targeting gRNA is because SCEPTURE only gives you an average score, while plotting the expressions of these cells together provide much more information on how these gRNAs change target gene expression. This is similar with scenarios when you analyze bulk RNA-seq, where you can just only show the average log fold change of two groups of samples (similar with just showing SCEPTURE score), or you plot the actual values of gene expressions between all samples in two groups. The latter will provide much more information for the readers to understand what the actual signals look like.

Nevertheless, this discrepancy should not prevent this paper to publish in Genome Biology.